# CRISPR/Cas9-engineered inducible gametocyte producer lines as a valuable tool for *Plasmodium falciparum* malaria transmission research

Sylwia D. Boltryk[1,2], Armin Passecker[1,2], Arne Alder[3,4,5], Eilidh Carrington [1,2], Marga van de Vegte-Bolmer[6], Geert-Jan van Gemert[6], Alex van der Starre[6], Hans-Peter Beck[1,2], Robert W. Sauerwein [6], Taco W. A. Kooij [6], Nicolas M. B. Brancucci[1,2], Nicholas I. Proellochs [6], Tim-Wolf Gilberger[3,4,5] & Till S. Voss [1,2✉]

The malaria parasite *Plasmodium falciparum* replicates inside erythrocytes in the blood of infected humans. During each replication cycle, a small proportion of parasites commits to sexual development and differentiates into gametocytes, which are essential for parasite transmission via the mosquito vector. Detailed molecular investigation of gametocyte biology and transmission has been hampered by difficulties in generating large numbers of these highly specialised cells. Here, we engineer *P. falciparum* NF54 inducible gametocyte producer (iGP) lines for the routine mass production of synchronous gametocytes via conditional overexpression of the sexual commitment factor GDV1. NF54/iGP lines consistently achieve sexual commitment rates of 75% and produce viable gametocytes that are transmissible by mosquitoes. We also demonstrate that further genetic engineering of NF54/iGP parasites is a valuable tool for the targeted exploration of gametocyte biology. In summary, we believe the iGP approach developed here will greatly expedite basic and applied malaria transmission stage research.

[1] Department of Medical Parasitology and Infection Biology, Swiss Tropical and Public Health Institute, Basel, Switzerland. [2] University of Basel, Basel, Switzerland. [3] Centre for Structural Systems Biology, Hamburg, Germany. [4] Bernhard Nocht Institute for Tropical Medicine, Hamburg, Germany. [5] University of Hamburg, Hamburg, Germany. [6] Department of Medical Microbiology, Radboudumc Center for Infectious Diseases, Radboud Institute for Molecular Life Sciences, Radboud University Medical Center, Nijmegen, The Netherlands. ✉email: till.voss@swisstph.ch

Malaria is a vector-borne infectious disease caused by protozoan parasites of the genus *Plasmodium*. Infections with *P. falciparum* are responsible for the vast majority of all malaria-related morbidity and mortality in humans. Over the last two decades, concerted intervention efforts targeting both the insect vector and the parasite led to a remarkable decline in malaria cases worldwide[1]. However, progress has come to a standstill in the past few years and in 2019, malaria was still accountable for 229 million clinical cases and 409,000 deaths, primarily in sub-Saharan Africa[2]. To further reduce the spread of malaria, intervention strategies will not only have to overcome the widespread resistance of mosquitoes and parasites to insecticides and first-line antimalarial drugs, respectively, but will also have to include efficient tools that interrupt parasite transmission from the human host to the mosquito vector[3,4].

People get infected with *P. falciparum* when infested female *Anopheles* spp. mosquitoes inject sporozoites into their skin. After reaching the liver via the bloodstream, sporozoites multiply within hepatocytes to release thousands of merozoites into circulation. Merozoites invade red blood cells (RBCs) and develop through the ring and trophozoite stage into a multinucleated schizont. After daughter cell formation, up to 32 merozoites egress from the infected RBC (iRBC) to invade and replicate inside new RBCs. Consecutive rounds of these intraerythrocytic developmental cycles (IDCs) are responsible for all disease symptoms and chronic infection. Importantly, however, during each replication cycle, a small proportion of schizonts produce sexually committed ring stage progeny that differentiate into either female or male gametocytes[5]. When taken up by a mosquito, terminally differentiated gametocytes egress from the iRBC and develop into gametes. The female gametocyte produces one macrogamete, while the male gametocyte undergoes three rapid rounds of genome replication followed by exflagellation of eight motile microgametes. After fertilisation, the zygote transforms into an ookinete that traverses the midgut epithelium to initiate sporogony, which ultimately renders the mosquito infectious to other humans. Hence, as the only forms of the parasite able to infect mosquitoes, gametocytes are an essential target for future antimalarial drugs aiming to prevent malaria transmission[4].

Gametocytogenesis entails two major processes: sexual commitment/conversion and sexual differentiation. Sexual commitment occurs during the IDC preceding gametocyte differentiation. This understanding was gained from experiments showing that all ring stage descendants derived from a single schizont have the same fate; they either all undergo another round of intracellular replication or they all differentiate into either female or male gametocytes[6,7]. In addition to this "next cycle sexual conversion (NCC)" process, recent studies reported "same cycle sexual conversion (SCC)" where ring stages directly commit to sexual development[8,9]. Irrespective of the NCC or SCC routes, sexual conversion is triggered by an epigenetic switch that activates expression of the master transcription factor AP2-G[10–13]. In asexual parasites, heterochromatin-dependent silencing of *ap2-g* prevents AP2-G expression[10,12–14]. In a small subset of trophozoites (NCC) or ring stages (SCC), however, the *ap2-g* locus gets activated by molecular mechanisms that are still largely unknown[9–13,15–17]. Gametocyte development 1 (GDV1), a nuclear protein essential for gametocytogenesis in *P. falciparum*[18], plays a key role in the NCC process[16]. GDV1 is specifically expressed in sexually committed trophozoites and schizonts, where it displaces heterochromatin protein 1 (HP1) from the *pfap2-g* locus, thereby licensing PfAP2-G expression[16]. Consistent with the epigenetic control mechanisms regulating *pfap2-g* expression, sexual commitment rates vary in response to environmental changes[5]. In particular, depletion of the host serum lipid lysophosphatidylcholine (LysoPC) triggers sexual commitment in trophozoites and this response is channelled

via induction of GDV1 and PfAP2-G expression[16,17]. Once expressed, PfAP2-G initiates a specific transcriptional program that drives sexual conversion and primes subsequent gametocyte differentiation[5]. While the PfAP2-G-dependent transcriptional changes in sexually committed schizonts are minor, a more pronounced gene expression signature emerges in the sexual ring stage progeny, where several dozen genes are specifically induced or repressed compared with asexual ring stages[10,12,13,15,16,19,20].

*P. falciparum* sexual ring stage parasites differentiate over a period of 10–12 days into transmission-competent mature female or male gametocytes. Sexual ring stages (day 1) develop into spherical stage I gametocytes (day 2) that continuously elongate into lemon-shaped stage II (day 4), D-shaped stage III (day 6), and spindle-shaped stage IV cells (day 8), before falciform-mature stage V gametocytes are formed (day 10+)[21,22]. These morphological transitions are linked to the gradual expansion of the inner membrane complex, an endomembrane system underlying the parasite plasma membrane, and the microtubule and actin cytoskeleton networks underneath that are disassembled again at the stage IV to V transition[23–27]. mRNA and protein expression profiling studies conducted mainly on late-stage gametocytes[28–32], but also on early stages[31,32] or across gametocyte maturation[33,34], identified hundreds of genes and proteins differentially expressed between gametocytes and asexual blood stage parasites, or between female and male gametocytes[35–38]. Some of these genes are known or predicted to be involved in biological processes that are altered in gametocytes compared with asexual parasites, such as host cell remodelling, energy and lipid metabolism, cytoskeleton organisation, transcriptional regulation, chromatin structure and translational repression[5,39]. During infection, stage I to IV gametocytes are sequestered away from circulation, primarily in the parenchyma of the bone marrow and spleen[40,41]. The mechanisms underlying gametocyte homing to and sequestration in these extravascular niches are only poorly understood. However, the high rigidity of stage I–IV gametocyte-infected RBCs[23,42,43], conferred by parasite-induced alterations of the RBC membrane and the underlying cytoskeletal networks[42,44–46], appears to play a primary role in gametocyte retention. Reversal of these modifications at the stage IV to V transition confers increased deformability[23,42–46], which is a likely prerequisite for the release of stage V gametocytes back into circulation[41,47]. Once in the bloodstream, stage V gametocytes circulate and remain competent for transmission to the mosquito for days/weeks[48]. The enormous transmission reservoir represented by the hundreds of millions of infected people in endemic areas, and the fact that almost all currently licensed antimalarial drugs, except primaquine, fail to kill mature gametocytes, poses major obstacles to malaria control and elimination efforts[4,49,50].

The discovery and development of new transmission-blocking drugs and vaccines requires a detailed functional and mechanistic understanding of the molecular processes underlying essential transmission stage biology and depends on robust experimental tools for routine basic and applied research. Both prerequisites are currently not met due to the challenges associated with the in vitro cultivation of gametocytes, particularly the generation of large numbers of synchronous gametocyte stages for experimental studies: (i) *P. falciparum* parasites show low sexual commitment rates (<10%), resulting in low gametocyte yields[5,41]; (ii) gametocytes are nonproliferative cells and thus are rapidly overgrown by asexual parasites; (iii) sexual commitment occurs during each consecutive IDC, which results in asynchronous gametocyte populations. Current approaches to increase sexual commitment rates in in vitro cultures are based on exposing parasites to poorly defined stress conditions such as high parasitemia and/or nutrient starvation (spent/conditioned medium)[22,51,52]. Several different protocols relying on this strategy exist and reach sexual commitment rates of 10–30%[53–59]. However, most of these protocols use cumbersome experimental workflows, rely on large culture

volumes, are expensive, difficult to reproduce, or produce asynchronous gametocyte populations. Protocols employing LysoPC- or choline-depleted minimal fatty acid medium[17] achieve sexual commitment rates in the range of 15–60%, depending on the strain used, but growth under these nutrient-restricted conditions reduces the number of sexually committed progeny[16,17,60,61]. A recent system based on an inducible promoter-swap approach at the *pfap2-g* locus triggers very high sexual conversion rates of up to 90%; however, gametocytes produced from this line are not infectious to mosquitoes and therefore unsuitable for research on late-stage gametocyte biology and gametocytocidal drug discovery[19].

Here, we show that the insertion of conditional GDV1 expression cassettes into the genome of the transmissible NF54 strain transforms these parasites into inducible gametocyte producer (iGP) lines suitable for the routine mass production of infectious gametocytes. Upon targeted induction of GDV1 expression, NF54/iGP parasites consistently achieve sexual conversion rates of 75% and generate synchronous gametocyte populations at high yield across all stages of gametocytogenesis. Importantly, NF54/iGP stage V gametocytes retain their capacity to infect female *Anopheles* mosquitoes and produce sporozoites able to infect human hepatocytes. Furthermore, by tagging the nuclear pore protein PfNUP313, we demonstrate that further genetic engineering of NF54/iGP lines is straightforward and a promising approach to study gametocyte biology.

## Results

### CRISPR/Cas9-based engineering of *P. falciparum* NF54 inducible gametocyte producer lines based on the targeted overexpression of GDV1. Using the FKBP destabilization domain (DD) system for controllable protein expression[62], we previously observed that conditional overexpression of plasmid-encoded GDV1-GFP-DD increases sexual conversion rates (SCRs) in the 3D7 reference strain[16]. Here, we exploited this finding in order to generate stable and marker-free inducible gametocyte producer (iGP) lines as tools to expedite basic and applied research on *P. falciparum* gametocyte biology and transmission.

As a proof of concept, we inserted a single GDV1-GFP-DD expression cassette into the dispensable *cg6* (*glp3*) locus (PF3D7_0709200)[63] in 3D7 parasites using a two-plasmid CRISPR/Cas9-based gene editing approach[16] (Fig. 1a and Supplementary Fig. 1a). To this end, we cotransfected the pHF_gC-*cg6* CRISPR/Cas9 plasmid (containing expression cassettes for the positive–negative selection marker human dihydrofolate reductase fused to yeast cytosine deaminase/uridyl phosphoribosyl transferase (hDHFR-yFCU), SpCas9, and a single-guide RNA targeting the *cg6* locus) and the pD_*cg6_cam-gdv1-gfp-dd* donor plasmid (containing the *gdv1-gfp-dd* transgene cassette flanked by *cg6* homology regions) into 3D7 wild-type (wt) parasites (Supplementary Fig. 1a). Characterisation of the resulting transgenic 3D7/iGP mother line and three clonal populations demonstrated that (i) the *gdv1-gfp-dd* transgene cassette was successfully inserted into the *cg6* locus (Fig. 1a, Supplementary Figs. 1b–d and Supplementary Note 1), and (ii) the addition of the stabilising ligand Shield-1 to the culture medium caused efficient induction of GDV1-GFP-DD expression in schizonts (Fig. 1b, Supplementary Fig. 2a and Supplementary Note 1). Importantly, IFAs probing for the expression of the gametocyte-specific marker Pfs16[64] revealed that up to 75% of parasites in the progeny of Shield-1-treated parasites [36–44 h post invasion (hpi), generation 2, day 2 of gametocytogenesis] represented early stage I gametocytes, compared with only 3–5% in the control populations cultured in the absence of Shield-1 (Fig. 1b–d, Supplementary Figs. 2b and 2c, and Supplementary Note 1).

Based on the encouraging results, we wanted to transfer the iGP system to NF54 parasites, the strain most widely used for the study of gametocyte biology and mosquito infection[65]. Unexpectedly, however, multiple attempts to insert the GDV1-GFP-DD expression cassette into the *cg6* locus in NF54 parasites were unsuccessful as we did not obtain viable parasites after multiple transfection experiments. While the reason for this failure is unknown, we suspected that the DD-dependent degradation of GDV1-GFP-DD may be less efficient in NF54 compared with 3D7 parasites, such that parasites harbouring the transgene cassette would commit to sexual differentiation at an increased rate and therefore display a slower proliferation rate. We therefore decided to employ the *glmS* riboswitch approach as an additional or alternative conditional expression system to regulate the ectopic expression of GDV1 in NF54 parasites. The *glmS* ribozyme element is usually placed within the untranslated region of mRNAs and mediates transcript degradation when activated by glucosamine (GlcN)[66,67]. We cloned two modified versions of the pD_*cg6_cam-gdv1-gfp-dd* donor plasmid by inserting a *glmS* ribozyme element directly downstream of the *gdv1-gfp-dd* (pD_*cg6_cam-gdv1-gfp-dd-glmS*) or *gdv1-gfp* coding sequence (pD_*cg6_cam-gdv1-gfp-glmS*), respectively. Cotransfection of the pBF_gC-*cg6* CRISPR/Cas9 plasmid (which carries the blasticidin deaminase (*bsd*) gene as a positive selection marker instead of h*dhfr*) with either of the donor plasmids allowed us to successfully select for transgenic NF54/iGP parasites in both instances (Fig. 2a and Supplementary Figs. 3a and 3b). PCRs on gDNA of the NF54/iGP1 line revealed complete disruption of the *cg6* locus and insertion of a single GDV1-GFP-DD-*glmS* expression cassette (Supplementary Fig. 3c). However, at least a subset of parasites in the population still contained the pBF_gC-*cg6* CRISPR/Cas9 and/or integrated donor plasmid concatamers. Treatment with 5-fluorocytosine (5-FC) efficiently eliminated parasites harbouring the pBF_gC-*cg6* plasmid but not those carrying integrated donor plasmid concatamers (Supplementary Fig. 3d). The NF54/iGP2 population was free of plasmids and showed complete disruption of the *cg6* locus through insertion of a single GDV1-GFP-*glmS* expression cassette (Fig. 2a and Supplementary Fig. 3e).

Induction of GDV1-GFP-DD expression in the NF54/iGP1 line through the simultaneous removal of GlcN and addition of Shield-1 (−GlcN/+Shield-1) produced progeny consisting of 74.8% (±4.6 SD) Pfs16-positive early stage I gametocytes (32–40 hpi in generation 2) compared with only 8.0% (±2.6 SD) in the control population (+GlcN/−Shield-1) (Fig. 2b). Likewise, induction of GDV1-GFP expression in the NF54/iGP2 line via removal of GlcN (−GlcN) delivered progeny consisting of 74.1% (±5.6 SD) early stage I gametocytes compared with only 5.7% (±1.3 SD) in parasites cultured in the presence of GlcN (+GlcN) (Fig. 2b). To monitor gametocyte maturation, the ring stage progeny of induced NF54/iGP1 and NF54/iGP2 parasites was cultured in medium containing 50 mM N-acetyl-D-glucosamine (GlcNAc) for six consecutive days (days 1–6 of gametocytogenesis) to eliminate asexual parasites[53,68] and was thereafter maintained under normal culture conditions, until day 12 of gametocyte development (Fig. 2c). Visual inspection of Giemsa-stained blood smears prepared daily from day 4 (stage II) onward revealed that these gametocytes differentiated in a highly synchronous manner into stage V gametocytes within 10–12 days (Fig. 2d and Supplementary Fig. S4). Notably, due to the high SCRs achieved, pure stage V gametocyte populations at 4–5% gametocytemia were consistently obtained (Fig. 2d). Hence, by applying the *glmS* riboswitch approach for the conditional overexpression of GDV1, either on its own or combined with the FKBP/DD system, we engineered two independent marker-free NF54/iGP lines that allow for the routine production of large numbers of pure and synchronous NF54 gametocytes across all stages of sexual differentiation.

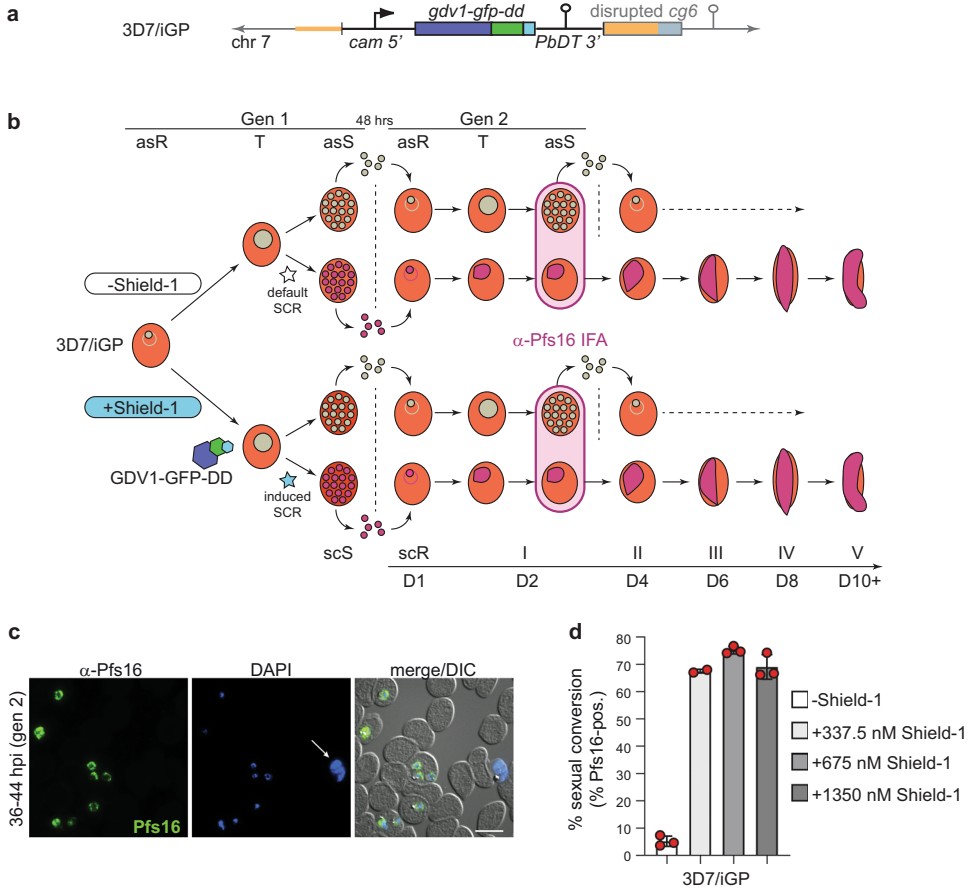

**Fig. 1 Description of the inducible gametocyte producer line 3D7/iGP. a** Schematic of the disrupted *cg6* locus carrying a single inducible GDV1-GFP-DD expression cassette. The 5′ and 3′ homology regions used for CRISPR/Cas9-based transgene insertion are shown in orange. **b** Schematic of the in vitro culture protocol used to quantify sexual commitment rates (SCRs). Synchronous 3D7/iGP ring stage parasites are split at 0–16 hpi and Shield-1 is added to one half of the population to trigger GDV1-GFP-DD expression. SCRs are quantified by determining the proportion of early stage I gametocytes in the total iRBC progeny 36–44 hpi (day 2 of gametocytogenesis) by α-Pfs16 IFAs combined with DAPI staining. Asexual parasites are depicted in grey, sexually committed parasites and gametocytes are depicted in purple. asS/scS, asexual/sexually committed schizont; asR/scR, asexual/sexually committed ring stage; T, trophozoite; I–V, gametocyte stages I–V; D1–D10, days 1–10 of gametocyte maturation. Gen 1/2, generation 1/2. **c** α-Pfs16 IFA images illustrating the high proportion of early gametocytes in the progeny of Shield-1-treated 3D7/iGP parasites. The white arrow highlights a Pfs16-negative schizont. Nuclei were stained with DAPI. DIC, differential interference contrast. Images are representative of three independent experiments. Scale bar, 10 μm. **d** Proportion of Pfs16-positive iRBCs (SCRs) in the progeny of 3D7/iGP treated with three different Shield-1 concentrations and the untreated control (−Shield-1) (mean ± SD, *n* = three biologically independent experiments; two experiments for parasites treated with 337.5 nM Shield-1). Closed circles represent data points for individual experiments (>156 DAPI-positive cells counted per experiment).

**NF54/iGP gametocytes produced via GDV1 overexpression display unaltered sex ratios and exflagellation rates**. To further validate NF54/iGP gametocytes as a valuable tool for experimental research, we generated clonal NF54/iGP1 and NF54/iGP2 populations by limiting dilution cloning. All clones were confirmed by PCR plasmid- and marker-free and carried a single GDV1-GFP-DD-*glmS* (NF54/iGP1) or GDV1-GFP-*glmS* (NF54/iGP2) expression module in the *cg6* locus (Supplementary Figs. 3c–e). In a preliminary experiment, induction of ectopic GDV1 expression triggered high SCRs that were somewhat lower compared with the corresponding mother lines, but still reached high values of 45–65% in all clonal lines (Supplementary Fig. 5a). Based on these results, we selected one clone each for further characterisation (NF54/iGP1_D8 and NF54/iGP2_E9). Sexual conversion assays performed in triplicate revealed that the background SCRs under noninducing conditions were low for both NF54/iGP1_D8 (+GlcN/−Shield-1) (8.2% ±3.5 s.d.) and NF54/iGP2_E9 (+GlcN) (9.0% ±2.8 s.d.), but slightly higher compared with NF54 wt parasites (2.0% ±0.5 s.d.) (Fig. 2e). This finding is not unexpected since SCRs

can vary strongly between strains and even isogenic clones[10]. Alternatively, it is conceivable that leaky ectopic expression of GDV1 in a small subset of parasites may be responsible for the slightly elevated background SCRs. When cultured under conditions that stabilise the *gdv1-gfp-dd* mRNA but not the GDV1-GFP-DD protein (−GlcN/−Shield-1), NF54/iGP1_D8 parasites still displayed low yet slightly increased SCRs of 13.3% (±5.5 s.d.) (Fig. 2e). This result shows that the DD-dependent degradation of GDV1-GFP-DD is functional in NF54/iGP1 parasites, but may indeed be less efficient compared with 3D7/iGP parasites that showed SCRs of only 3–5% in the absence of Shield-1 (Fig. 1a and Supplementary Figs. 2c and 2e). Importantly, however, upon induction of GDV1-GFP-DD and GDV1-GFP expression, respectively, both clones showed massively increased SCRs reaching 62.8% (±1.7 s.d.) for NF54/iGP1_D8 (−GlcN/+Shield-1) and 72.9% (±3.7 s.d.) for NF54/iGP2_E9 (−GlcN) (Fig. 2e). Consistent with these functional readouts, Western blot analysis of schizont samples confirmed the ectopic expression of GDV1 only under inducing conditions in both lines (Fig. 2f and Supplementary Fig. 5b). Expression of

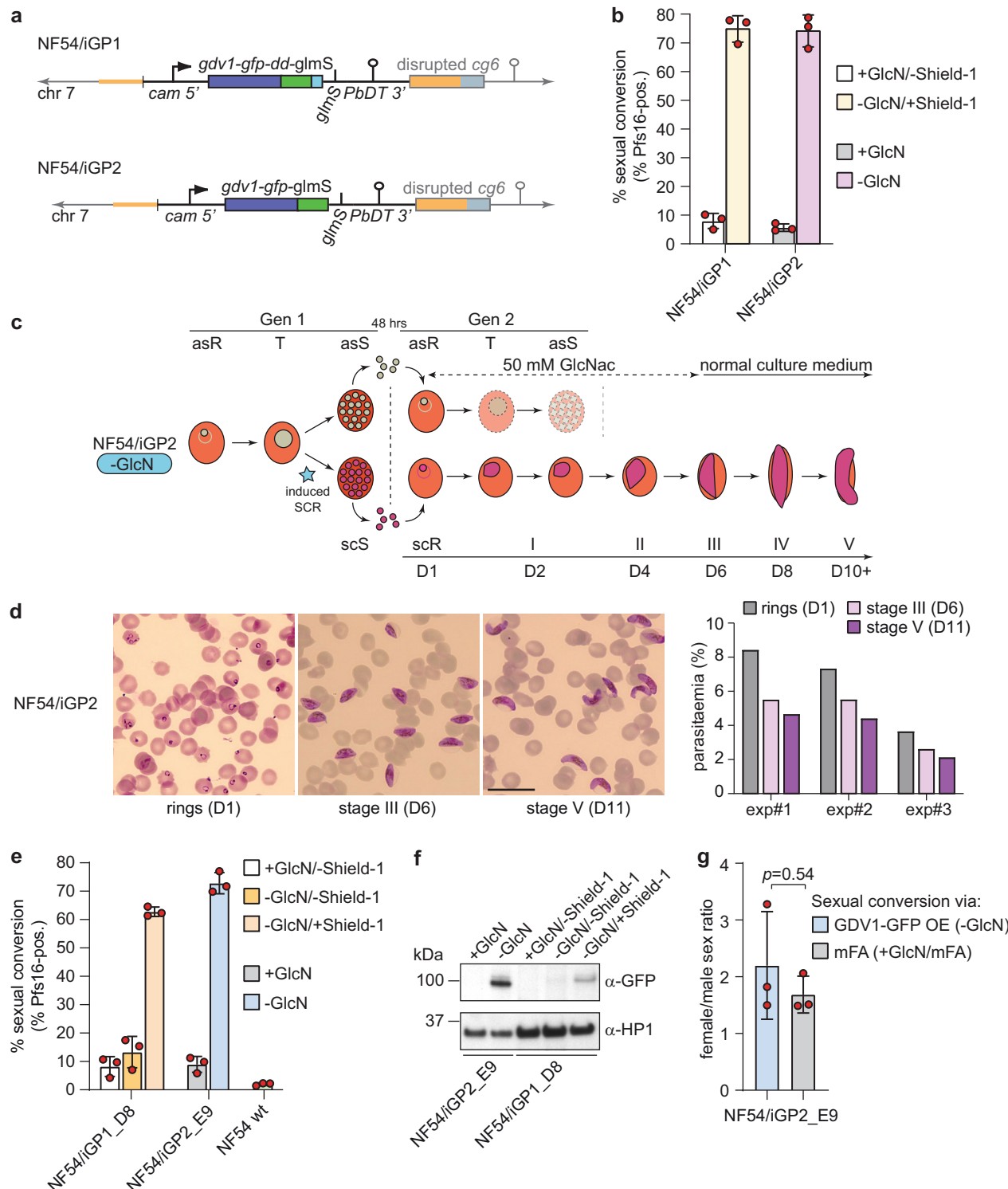

GDV1-GFP-DD in NF54/iGP1_D8 parasites (−GlcN/+Shield-1) was substantially weaker compared with GDV1-GFP in NF54/iGP2_E9 parasites (−GlcN), suggesting that GDV1-GFP-DD expression is only moderately stabilised by Shield-1 (Fig. 2f and Supplementary Fig. 5b). We also tested whether ectopic expression of GDV1 occurs in gametocytes cultured in medium lacking both GlcN and Shield-1. Whereas GDV1-GFP-DD was not expressed in NF54/iGP1_D8 gametocytes as expected, GDV1-GFP expression was detectable in NF54/iGP2_E9 gametocytes, but at much lower levels compared with asexual

parasites and completely abolished by addition of GlcN (Supplementary Fig. 5c).

To confirm that induction of sexual commitment via ectopic GDV1 expression has no influence on gametocyte sex ratios, we performed IFAs on stage V gametocytes (day 10) using antibodies against the female-specific protein Pfg377[69] (Supplementary Fig. 5d). We observed no significant difference in female/male sex ratios between NF54/iGP2_E9 gametocytes obtained via GDV1-GFP overexpression (−GlcN; 2.2 ±0.8 s.d.) or via induction of sexual commitment through LysoPC-/choline-depleted minimal

**Fig. 2 Description of the inducible gametocyte producer lines NF54/iGP1 and NF54/iGP2. a** Schematics of the disrupted *cg6* locus carrying a single inducible GDV1-GFP-DD-*glmS* or GDV1-GFP-*glmS* expression cassette in NF54/iGP1 or NF54/iGP2, respectively. The 5′ and 3′ homology regions used for CRISPR/Cas9-based transgene insertion are shown in orange. **b** Proportion of Pfs16-positive iRBCs (SCRs) in the progeny of NF54/iGP1 and NF54/iGP2 cultured under noninducing or inducing conditions (mean ± SD, *n* = three biologically independent experiments). Closed circles represent data points for individual experiments (≥391 DAPI-positive cells counted per experiment). **c** Schematic of the in vitro culture protocol used to obtain pure NF54/iGP2 stage V gametocyte populations. GlcN is removed from the culture medium of synchronous ring stage parasites at 0–16 hpi to trigger expression of GDV1-GFP. After schizont rupture and merozoite invasion, gametocyte maturation proceeds for >10 days. Asexual parasites are depicted in grey, sexually committed parasites and gametocytes are depicted in purple. asS/scS, asexual/sexually committed schizont; asR/scR, asexual/sexually committed ring stage; T, trophozoite; I–V, gametocyte stages I–V; D1–D10, days 1–10 of gametocyte maturation. Gen 1/2, generation 1/2. **d** Images of Giemsa-stained NF54/iGP2 gametocyte cultures, acquired on day 1 (asexual/sexually committed ring stages), day 6 (stage III gametocytes), and day 11 (stage V gametocytes). Images are representative of three independent experiments. Scale bar, 20 μm. Parasitemias determined from three independent induction experiments are shown on the right (≥2068 RBCs counted per experiment). **e** Proportion of Pfs16-positive iRBCs (SCRs) in the progeny of NF54/iGP1_D8 and NF54/iGP2_E9 parasites cultured under noninducing or inducing conditions and of NF54 wt control parasites (mean ± SD, *n* = three biologically independent experiments). Closed circles represent data points for individual experiments (≥143 DAPI-positive cells counted per experiment). **f** Western blot showing expression of GDV1-GFP (MW=99.1 kDa) and GDV1-GFP-DD (MW=111.3 kDa) in NF54/iGP2_E9 and NF54/iGP1_D8 schizonts (34–42 hpi), respectively. PfHP1 (MW = 31 kDa) served as a control to compare the relative numbers of nuclei loaded per lane. The results are representative of two independent experiments. **g** Sex ratios of NF54/iGP2_E9 stage V gametocytes obtained via GDV1-GFP overexpression (−GlcN) or via induction of sexual commitment using mFA medium (+GlcN/mFA), as quantified from α-Pfg377 IFAs (mean ± SD, *n* = three biologically independent experiments). Closed circles represent data points for individual experiments (≥192 gametocytes scored per experiment). Sex ratios were compared using a paired two-tailed Student's *t* test (*p* value indicated above the graph).

---

fatty acid medium (+GlcN/mFA; 1.7 ±0.3 s.d.) (*p* = 0.54; paired two-tailed Student's *t* test) (Fig. 2g and Supplementary Fig. 6a). The same result was obtained from a single experiment performed using the NF54/iGP1_D8 clone (Supplementary Fig. 6b). Together, these results show that GDV1 overexpression has no effect on gametocyte sex ratios, which in turn suggests that GDV1 is not involved in the hitherto unknown sex determination pathway in *P. falciparum*. Finally, as a proxy to assess gametocyte viability, we quantified male stage V gametocyte exflagellation on days 10, 13, and 14 of gametocytogenesis and demonstrate that both NF54/iGP1_D8 and NF54/iGP2_E9 gametocytes exflagellated as efficiently as NF54 wt gametocytes, and similar results were obtained for the corresponding NF54/iGP mother lines (Supplementary Figs. 6c and 6d).

**NF54/iGP1 and NF54/iGP2 gametocytes are infectious to mosquitoes and produce viable sporozoites that infect human hepatocytes.** To test if NF54/iGP gametocytes retained their capacity to undergo fertilisation and further life cycle progression in the mosquito vector, we fed stage V gametocytes to female *Anopheles stephensi* mosquitoes using Standard Membrane Ffeeding Aassays (SMFAs)[65,70]. To this end, we triggered sexual conversion in clones NF54/iGP1_D8 (−GlcN/+Shield-1) and NF54/iGP2_E9 (−GlcN) as outlined in Fig. 2c, and maintained the ring stage progeny (day 1 of gametocytogenesis) in culture for 14 days with daily medium changes to ensure complete differentiation into mature stage V gametocytes. For each of the two populations, separate SMFAs were performed on days 10, 13, and 14 of gametocytogenesis, and the entire experiment was repeated with a second batch of independently produced gametocytes. Eight days after the feeds, 20 mosquitoes each were dissected and midgut oocysts counted by microscopy. Both NF54/iGP clones infected mosquitoes as efficiently as NF54 wt gametocytes on each of the three days of feeding, with 87.5–97.5% (NF54/iGP1_D8), 92.5–100% (NF54/iGP2_E9), and 90–100% (NF54 wt) of mosquitoes carrying midgut oocysts (Fig. 3a). Mosquitoes infected with NF54/iGP1_D8 gametocytes consistently developed about half as many oocysts compared with the NF54 wt control, and for both of these lines the highest oocyst numbers were observed in mosquitoes fed with day 13 gametocytes (median=13 oocysts/mosquito for NF54/iGP1_D8; median=27 oocysts/mosquito for NF54 wt) (Fig. 3a). Interestingly, NF54/iGP2_E9 gametocytes achieved even higher oocyst intensities compared with NF54 wt gametocytes and were most

infectious on day 13 (median = 37 oocysts/mosquito) and day 14 (median = 47 oocysts/mosquito) (Fig. 3a). Gametocytes of the NF54/iGP1 and NF54/iGP2 mother lines were also infectious to mosquitoes. While infections with NF54/iGP1 gametocytes resulted in infection rates and oocyst intensities similar to those obtained with the clonal NF54/iGP1_D8 line, NF54/iGP2 gametocytes infected fewer mosquitoes and developed fewer oocysts on all three feeding days (Supplementary Fig. 7a).

To assess sporozoite production and viability, salivary gland sporozoites were isolated on day 17 post infection from mosquitoes infected with day 14 gametocytes. We observed higher salivary gland sporozoite intensities in mosquitoes infected with clones NF54/iGP1_D8 (mean=69,510 sporozoites/mosquito) and NF54/iGP2_E9 (mean=101,692 sporozoites/mosquito) (Fig. 3b), as well as the NF54/iGP1 mother line (mean=69,657 sporozoites/mosquito) (Supplementary Fig. 7b), compared with the NF54 wt control (mean=24,676 sporozoites/mosquito) (Fig. 3b). Due to the low number of midgut oocysts in mosquitoes infected with NF54/iGP2 mother line gametocytes, strongly reduced sporozoite intensities were observed in these infections (mean=6,265 sporozoites/mosquito) (Supplementary Fig. 7b). When normalised to the mean oocyst intensities, the numbers of sporozoites per oocyst were about twice as high for NF54/iGP1_D8 (mean=4065 sporozoites/oocyst) and NF54/iGP1 (mean=3440 sporozoites/oocyst) compared with NF54/iGP2_E9 (mean=1930 sporozoites/oocyst), NF54/iGP2 (mean=1989 sporozoites/oocyst), and NF54 wt (mean=1552 sporozoites/oocyst) (Fig. 3b and Supplementary Fig. 7b). To assess sporozoite infectivity, we conducted a hepatocyte infection assay using primary human hepatocytes. Confocal fluorescence microscopy analysis of cells stained with antibodies against PfHSP70 (parasite cytosol) and PfEXP2 (parasitophorous vacuolar membrane) showed that NF54/iGP1_D8, NF54/iGP2_E9, and NF54/iGP1 sporozoites successfully invaded hepatocytes and underwent intrahepatocytic maturation comparable with the NF54 wt control, at least until day 5 post infection (note that the low number of sporozoites isolated from NF54/iGP2-infected mosquitoes were insufficient to perform hepatocyte-invasion assays) (Fig. 3c and Supplementary Fig. 7c). Due to the low infection rates observed for all lines in this particular experiment, however, we were unable to quantify hepatocyte infection. Noteworthy, for both clones, ectopic GDV1 expression was undetectable in oocysts and salivary gland sporozoites by live-cell fluorescence microscopy, or in liver-stage parasites by IFAs using α-GFP antibodies (Fig. 3c and Supplementary Fig. 7c).

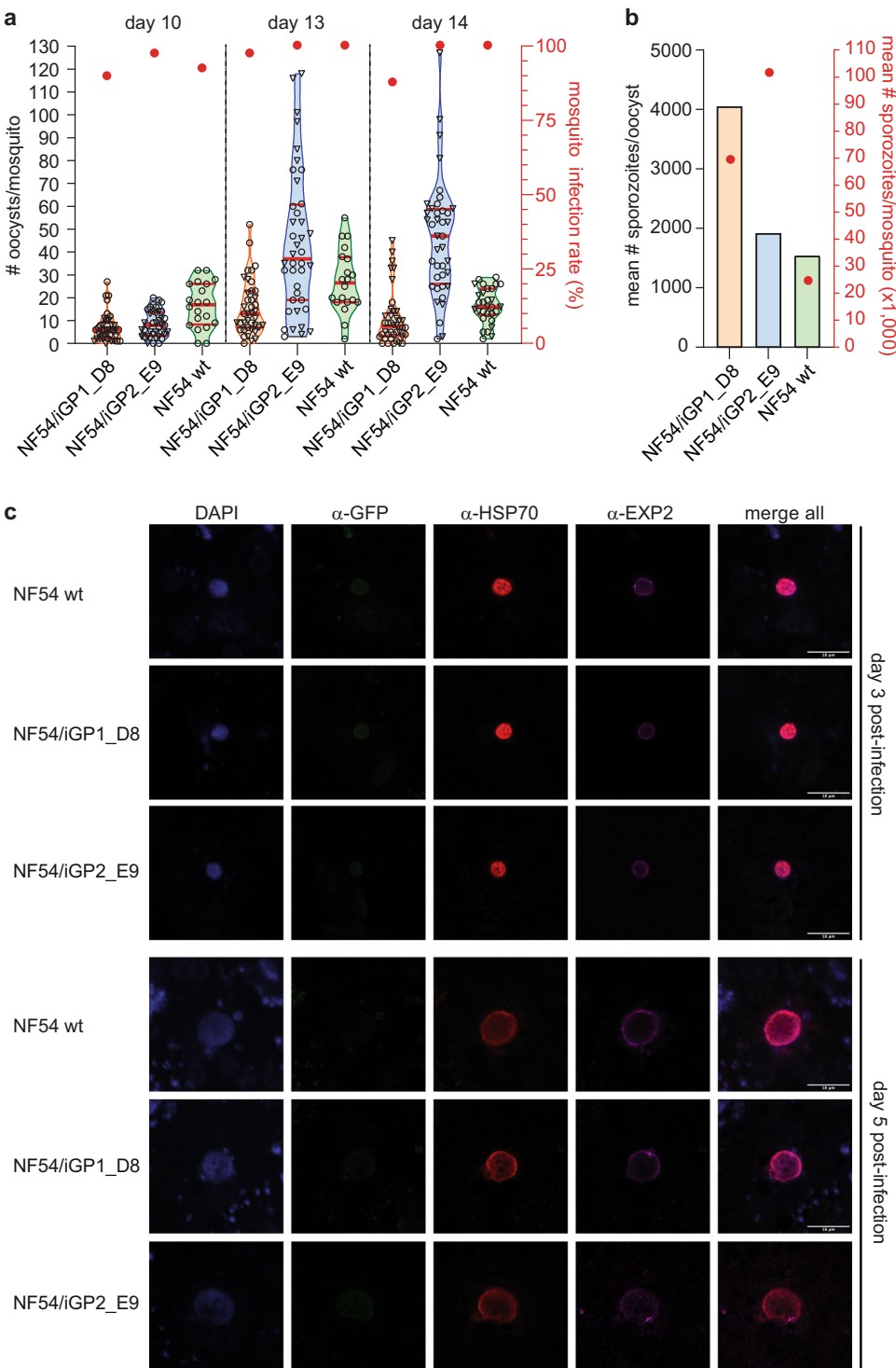

Together, these results demonstrate that the NF54/iGP1_D8 and NF54/iGP2_E9 clones, and the NF54/iGP1 mother line, complete their life cycle in the mosquito vector as efficiently as NF54 wt parasites and produce sporozoites able to infect and mature within human liver cells.

**Further genetic manipulation of NF54/iGP parasites for the targeted investigation of gametocyte biology.** To explore NF54/iGP parasites as a tool to study specific aspects of gametocyte biology, we engineered NF54/iGP2 parasites expressing a mScarlet-tagged version of the nuclear pore protein PfNUP313

(Supplementary Fig. 8). Nuclear pores are large macromolecular complexes that are embedded in the nuclear envelope and consist of approximately 30 different nucleoporins (NUPs) (each present in multiple copies)[71]. Nuclear pore complexes (NPCs) act as essential gateways for molecular transport into and out of the nucleus and play additional crucial roles in the regulation of gene expression and genome organisation[71,72].

In malaria parasites, nuclear pores have hardly been studied and only six NUPs have been identified so far[73–77], of which four have been localised in *P. falciparum* asexual blood stage parasites [PfNUP116 (PF3D7_1473700)[74], PfNUP221/PfNUP100

**Fig. 3 NF54/iGP1_D8 and NF54/iGP2_E9 gametocytes complete their life cycle in the mosquito vector and produce infectious sporozoites. a** NF54/iGP1_D8 (orange), NF54/iGP2_E9 (blue), and NF54 wt control stage V gametocytes (green) were fed to female *Anopheles stephensi* mosquitoes on day 10, 13, and 14 of gametocytogenesis in two independent SMFA experiments. The violin plots show the distribution of the number of oocysts detected in each of the 20 mosquitoes dissected per feed, with open circles and triangles representing data from SMFA replicates 1 and 2, respectively (left *y* axis). The median (thick red line) and upper and lower quartiles (thin red lines) are indicated. Closed red circles represent the mean oocyst prevalence (number of infected mosquitoes) determined for each of the two replicate feeds (right *y* axis). NF54 wt day 10 and day 13 gametocytes were only included in SMFA replicate 1, and for SMFA replicate 2, only 10 mosquitoes infected with NF54 wt day 14 gametocytes have been dissected. **b** Mean number of salivary gland sporozoites per oocyst (left *y* axis) and per mosquito (closed red circles; right *y* axis) 17 days after infection with NF54/iGP1_D8 (orange), NF54/iGP2_E9 (blue), and NF54 wt control day 14 gametocytes (green) (SMFA replicate 2 data). Values represent the results from a single experiment (≥26 mosquitoes dissected per infected batch). **c** Confocal microscopy IFA images showing intracellular parasites after infection of primary human hepatocytes with NF54/iGP1_D8, NF54/iGP2_E9, and NF54 wt control sporozoites. Parasites were stained with α-PfHSP70 (cytosol; red) and α-PfEXP2 antibodies (parasitophorous vacuole membrane; purple). α-GFP antibodies were used to test for potential ectopic expression of GDV1 in liver stages. Nuclei were stained with DAPI. Images are representative of a single experiment. Scale bar, 18 μm.

(PF3D7_0905100)[77], PfNUP313 (PF3D7_1446500)[73], and PfSEC13 (PF3D7_1230700)[76]]. We were interested in visualising nuclear pores in gametocytes for two reasons. First, the number and distribution of NPCs within the nuclear envelope of asexual parasites changes dramatically as they progress through the IDC[74,78,79]. The functional relevance of these dynamic NPC expression patterns is currently unknown, but may be linked to regulatory strategies of stage-specific gene expression and/or the coordination of nuclear segregation during schizogony[78]. We therefore wished to learn how NPC abundance and localisation compares in differentiating gametocytes. Second, early studies investigating gametocyte morphology at the ultrastructural level suggested a pronounced nuclear dimorphism between female and male gametocytes[26,80], but this has not been further explored in any great detail. To revisit this intriguing observation, we wanted to visualise the outer confines of the parasite nucleus throughout gametocytogenesis by fluorescence microscopy. Since *P. falciparum* lacks homologues of known nuclear envelope-associated proteins such as lamin, we used the NPC as a surrogate marker for the nuclear membrane.

To obtain NF54/iGP2_NUP313-mSc parasites, we cotransfected the pBF_gC-*nup313* CRISPR/Cas9 plasmid and the pD_*nup313-mScarlet* donor plasmid into the NF54/iGP2 line (Supplementary Fig. 8). Transgenic parasites were successfully selected on BSD-S-HCl and PCRs on gDNA confirmed correct editing of the *nup313* gene and absence of parasites carrying the wt locus (Supplementary Fig. 8). Validation of the NF54/iGP2_NUP313-mSc line by live-cell fluorescence imaging verified the expected localisation of nuclear pores across the different IDC stages as determined in previous studies[74,78,79] (Fig. 4a). PfNUP313-mScarlet localised to a single region adjacent to the DAPI-stained area in merozoites and early ring stages. The increased number and even distribution of NPCs within the nuclear envelope in trophozoites and early schizonts was reflected by a circular perinuclear pattern of PfNUP313-mScarlet foci directly adjoining the genetic material. In late schizonts, the number of PfNUP313-mScarlet signals decreased again to one or two per nucleus. To assess PfNUP313-mScarlet localisation in gametocytes, we induced sexual commitment in NF54/iGP2_NUP313-mSc parasites according to the protocol outlined in Fig. 2c and performed live-cell fluorescence imaging for all five gametocyte stages. As shown in Fig. 4b, PfNUP313-mScarlet signals were abundant and surrounded the often elongated area of nuclear DNA in a dot-like fashion in all five gametocyte stages, similar to the localisation pattern observed in trophozoites (Fig. 4a). However, in stage II to V gametocytes, the NPC signals frequently stretched away from bulk chromatin, suggestive of rounded expansions (pink arrowheads) and narrow lateral extensions (yellow arrowheads) of the nuclear envelope that sometimes reached close to the cellular poles (Figs. 4b and 5,

Supplementary Fig. 9). Primarily in late-stage gametocytes, we also observed nuclei with two clearly separate PfNUP313-delineated regions, of which either both (white arrowheads) or only one (blue arrowhead) contained genetic material detectable by Hoechst staining (Fig. 5). In summary, our findings show that gametocyte nuclei contain a similar or even higher density of NPCs compared with trophozoites, undergo profound changes in nuclear shape and size, and often contain regions of low chromatin density.

## Discussion

Here, we engineered robust inducible gametocyte producer (iGP) lines that will allow investigating gametocyte biology with a similar level of routine, depth, and detail hitherto reserved only for the study of asexual blood stage parasites. We achieved this by inserting a single conditional GDV1 expression cassette into the genomes of the 3D7 and NF54 strains. In these parasites, a single pulse of ectopic GDV1 overexpression in trophozoites and schizonts is sufficient to induce sexual commitment in a high proportion of parasites. Importantly, this approach mimics the natural process of GDV1-dependent sexual conversion. GDV1 is a specific and essential activator of PfAP2-G expression, and induction of endogenous GDV1 expression in trophozoites and schizonts is part of the inherent pathway of sexual commitment[16,18,81]. Furthermore, we have previously shown that the temporary activation of ectopic GDV1 expression triggers the typical PfAP2-G-dependent transcriptional cascade of sexual conversion and early gametocyte differentiation and has no effect on the transcription of other genes[16].

The iGP lines described in this study overcome the limitations of current protocols used for the bulk preparation of gametocytes. First, induction of sexual commitment is entirely independent of cumbersome and unreliable culture-handling protocols and only requires adding Shield-1 and/or removing GlcN from the culture medium to induce ectopic GDV1 expression. Second, the targeted induction of ectopic GDV1 expression triggers consistent sexual commitment rates of up to 75%, which allows generating large numbers of gametocytes even from small culture volumes. Third, sexual commitment is induced in a synchronous population of asexual blood stage parasites, which leads to synchronous sexual differentiation in the progeny and therefore facilitates preparing stage-specific gametocyte populations across gametocyte maturation. Fourth, gametocyte synchronicity and yield are highly reproducible between experiments; starting with approximately $0.75 \times 10^7$ ring-stage parasites in the commitment cycle (5% haematocrit, 1.5% parasitemia) routinely delivers up to $2 \times 10^8$ stage V gametocytes per 10 ml of culture in a single induction experiment (Fig. 6), which increases gametocyte yields by at least one order of magnitude compared with previous reports[53,58,82]. Because gametocyte production can be induced once every second day from a synchronous asexual feeder culture, the iGP lines

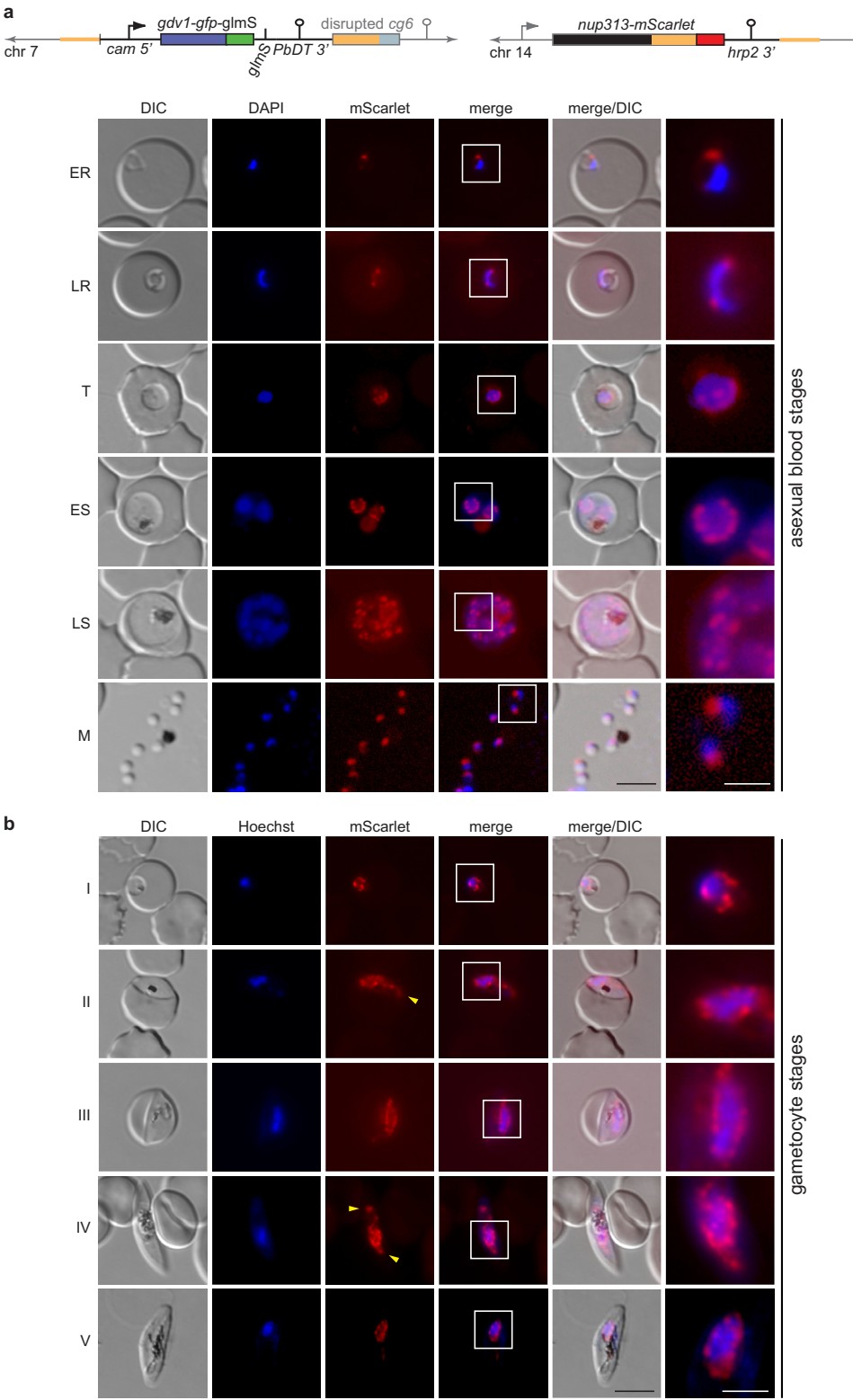

provide a constant rich source of synchronous gametocyte populations for experimental studies.

Similar to the recently developed 3D7/E5ind line that uses a conditional promoter-swap approach to induce PfAP2-G expression[19], the 3D7/iGP line produces exflagellation-defective male stage V gametocytes and is therefore of limited value for the investigation of late-stage gametocyte biology and gametocytocidal drug discovery. However, 3D7/iGP parasites still provide a useful tool to dissect the molecular events linked to sexual conversion and early gametocyte differentiation. The 3D7/E5ind line is superior in this regard as it displays virtually zero background sexual commitment and achieves sexual conversion rates of up to 90% upon induction of PfAP2-G expression, which allows identifying molecular signatures of sexually committed parasites with high precision[19]. 3D7/iGP parasites will be useful to complement such studies and to investigate the mechanisms involved in GDV1-dependent activation of *pfap2-g*.

**Fig. 4 Visualisation of nuclear pore distribution in NF54/iGP2_NUP313-mSc asexual blood stage parasites and gametocytes. a** Schematic maps of the disrupted *cg6* locus carrying a single inducible GDV1-GFP-glmS expression cassette and the tagged *nup313* locus in the double-transgenic NF54/iGP2_NUP313-mSc line are shown on top. The 5′ and 3′ homology regions used for CRISPR/Cas9-based genome editing are shown in orange. Live-cell fluorescence microscopy images showing the localisation of NUP313-mScarlet (red) in asexual blood stage parasites. ER/LR, early/late ring stage; T, trophozoite; ES/LS, early/late schizont; M, merozoite. DIC, differential interference contrast. Nuclei were stained with DAPI. Images are representative of three biologically independent experiments. Scale bar, 5 μm. White frames refer to the magnified view presented in the rightmost images (scale bar, 2 μm). **b** Live-cell fluorescence microscopy images showing the localisation of NUP313-mScarlet (red) in stage I to V gametocytes. Lateral extensions of the nucleus away from Hoechst-stained bulk chromatin are highlighted by yellow arrowheads. I–V, stage I to V gametocytes. DIC, differential interference contrast. Nuclei were stained with Hoechst (blue). Images are representative of four biologically independent experiments. Scale bar, 5 μm. White frames refer to the magnified view presented in the rightmost images (scale bar, 2 μm).

In contrast to the above-mentioned cell lines, NF54/iGP gametocytes are infectious to mosquitoes. Our comprehensive characterisation of NF54/iGP1_D8 and NF54/iGP2_E9 validated these clonal lines as most promising tools for future research. Both clones infect mosquitoes and produce oocysts and infectious salivary gland sporozoites as efficiently as NF54 wt parasites. Furthermore, the mosquito infection rates and oocyst/sporozoite intensities achieved in this study are consistent with SMFA outcomes usually obtained with NF54 wt gametocytes[83–85]. Moreover, NF54/iGP1_D8 and NF54/iGP2_E9 parasites are confirmed plasmid- and marker-free and therefore allow further rounds of genetic manipulation using any of the three drug resistance markers routinely used for selection of transgenic parasites (h*dhfr*, *bsd*, yeast dihydroorotate dehydrogenase). Manipulating genes in NF54/iGP parasites provides the great advantage that the study of gene and protein function by phenotypic, cell biological, biochemical, or structural analyses can routinely be performed not only in asexual blood stage parasites but readily also in a large number of isogenic gametocytes. Owing to these favourable properties, NF54/iGP1_D8 and NF54/iGP2_E9 parasites offer numerous opportunities for basic and applied malaria transmission stage research (Fig. 6; see also Supplementary Table 2 for a comparison of both clones and recommendations for their use in future research). For instance, NF54/iGP parasites will facilitate the targeted dissection of molecular mechanisms underlying the specific biology of gametocytes. They will also lend themselves for time-resolved high-throughput profiling experiments to generate comprehensive transcriptomics, (phospho-)proteomics, epigenomics and metabolomics reference and experimental datasets for each stage of gametocyte development. Indeed, NF54/iGP2 parasites have recently been used for in-depth comparative complexome profiling experiments, revealing crucial insight into the differential abundance and composition of mitochondrial respiratory chain complexes between asexual blood stage parasites and gametocytes[86]. Importantly, NF54/iGP parasites will also simplify and streamline high-throughput drug screening campaigns to identify gametocytocidal compounds and support the preclinical development of transmission-blocking drugs and vaccines. NF54/iGP gametocytes may further provide a reliable resource for research on mosquito and liver stage parasites and for the optimisation of protocols aiming to produce zygotes, ookinetes, oocysts, and sporozoites in vitro[87–89] (Fig. 6). Transferring the iGP approach to other *P. falciparum* strains could facilitate the systematic analysis of potential strain-dependent differences in gametocyte biology, sensitivity to antimalarial drugs, infectiousness to mosquitoes, or the capacity to undergo cross-fertilisation.

We illustrated the feasibility of subjecting NF54/iGP parasites to a second round of genetic engineering by tagging the nuclear pore component PfNUP313 in the NF54/iGP2 line. Our fluorescence microscopy data demonstrate marked differences in NPC abundance and distribution in the nuclear envelope, as well as in nuclear morphology between asexual blood stage parasites and gametocytes. The results obtained from asexual parasites recapitulate previous findings obtained by high-resolution microscopy[74,78,79]. These studies showed that merozoites and ring stage parasites possess only three to seven nuclear pores that are closely clustered in one region of the nuclear membrane. As ring stage parasites develop into trophozoites, the number of NPCs increases and up to 60 NPCs are evenly distributed throughout the nuclear envelope. During schizogony, the number of pores per nucleus decreases with increasing numbers of nuclei formed, such that in late schizonts, each nucleus of the developing merozoites again possesses only a small number of clustered nuclear pores[74,78,79]. In contrast, we found that gametocyte nuclei contain a stable and relatively high number of evenly distributed NPCs throughout all stages of gametocyte development, similar to what is observed for trophozoite nuclei. These observations are consistent with the open chromatin conformation and high transcriptional activity observed in both life cycle stages[90–92] and are suggestive of a high demand for nucleocytoplasmic shuttling during periods of growth. Importantly, because NPCs serve as a surrogate marker delineating the nuclear envelope, our results also allow us to conclude that the nucleus in stage II to V gametocytes (i) undergoes marked morphological transformations reflected in narrow lateral extensions along the longitudinal axis of the gametocyte or rounded expansions of the nuclear envelope; and (ii) is frequently considerably larger than what might be extrapolated from the Hoechst-stained bulk of genetic material, in line with observations made in *P. berghei* gametocytes[75]. In some late-stage gametocytes, we even observed two distinct NPC-demarcated regions, of which one or both stained positive with Hoechst. Together, these data suggest that nuclear reorganisation and functional genome compartmentalisation plays an important role in transcriptional reprogramming during gametocytogenesis and/or in the preparation for genome endoreplication and fertilisation in male and female stage V gametocytes, respectively. Interestingly, the results obtained from transmission electron microscopy studies suggested that the nuclei of female and male gametocytes differ markedly both in size and shape, with the female nucleus being generally oval-shaped and comparably small and the male nucleus appearing substantially larger with a lobular shape and distending toward the poles of the gametocyte[26,80]. It is therefore tempting to speculate that the cells containing enlarged and irregularly shaped nuclei detected in our study may primarily represent male gametocytes. However, although differential haemozoin crystal distribution has been proposed as a diagnostic feature to distinguish between the two sexes (dispersed in males, clustered in females)[21,26], we observed ambiguous pigment patterns and irregular nuclear shapes in gametocytes with clustered or dispersed haemozoin distribution alike (Figs. 4b and 5 and Supplementary Fig. 9), suggesting that the nucleus undergoes marked transformations in both male and female gametocytes. Notwithstanding this uncertainty, our analysis of NF54/iGP2_NUP313-mSc parasites revealed new insight into the intriguing morphological features of gametocyte nuclei based on standard fluorescence microscopy and provides an excellent starting point for further detailed cytological and functional investigations of nuclear biology in *P. falciparum* transmission stages.

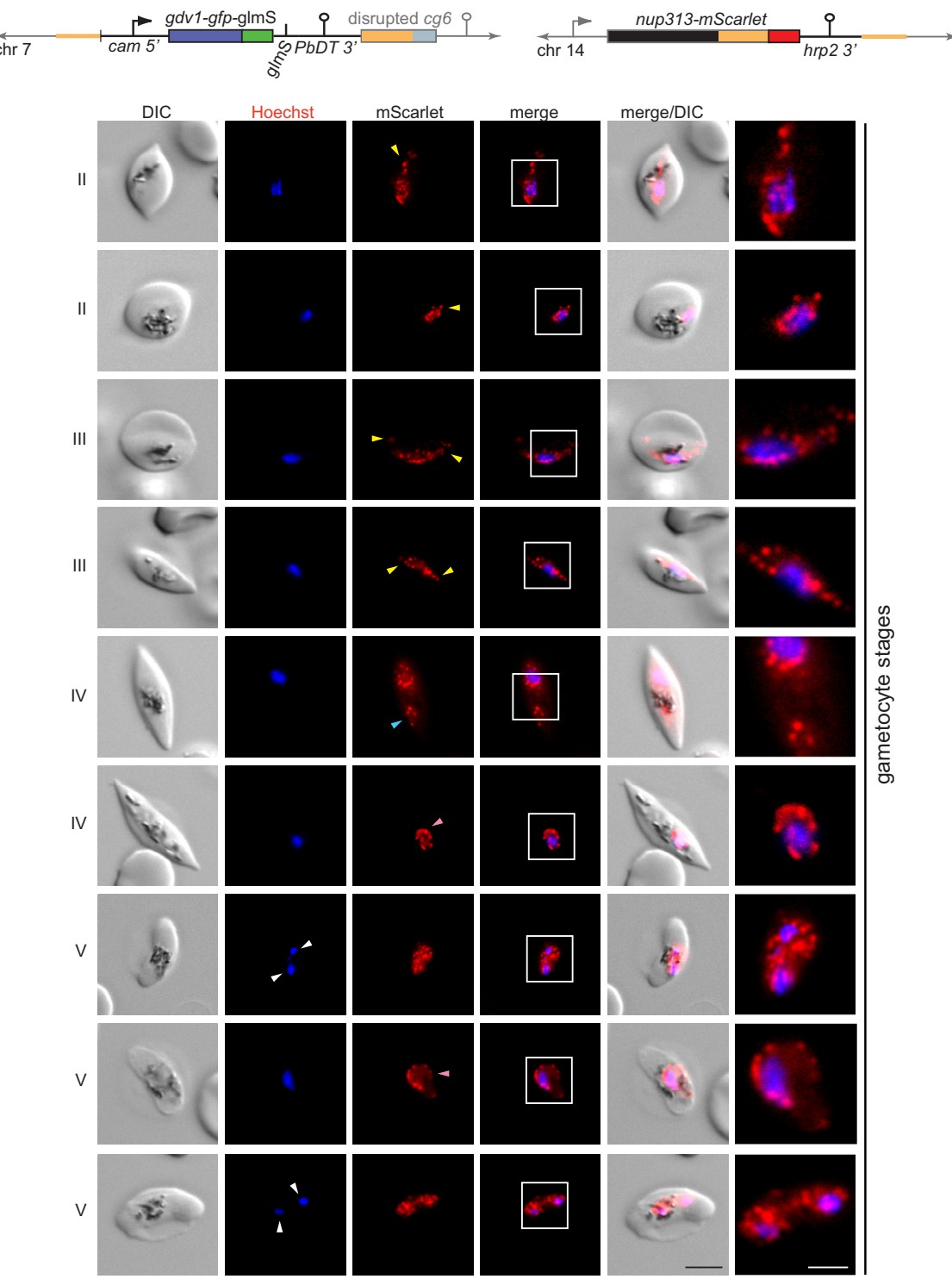

**Fig. 5 Nuclei in stage II–V gametocytes undergo marked morphological transformations.** Schematic maps of the disrupted *cg6* locus carrying a single inducible GDV1-GFP-glmS expression cassette and the tagged *nup313* locus in double-transgenic NF54/iGP2_NUP313-mSc line are shown on top. The 5′ and 3′ homology regions used for CRISPR/Cas9-based genome editing are shown in orange. Live-cell fluorescence microscopy images showing the localisation of NUP313-mScarlet (red) in stage II to V gametocytes. Lateral extensions (yellow arrowheads) or rounded expansions (pink arrowheads) of the nucleus away from Hoechst-stained bulk chromatin and separate NUP313-mScarlet-delineated regions enclosing (white arrowheads) or devoid of Hoechst-stained bulk chromatin (blue arrowhead) are highlighted. II–V, stage II to V gametocytes. DIC, differential interference contrast. Nuclei were stained with Hoechst. Images are representative of four biologically independent experiments. Scale bar, 5 μm. White frames refer to the magnified view presented in the rightmost images (scale bar, 2 μm).

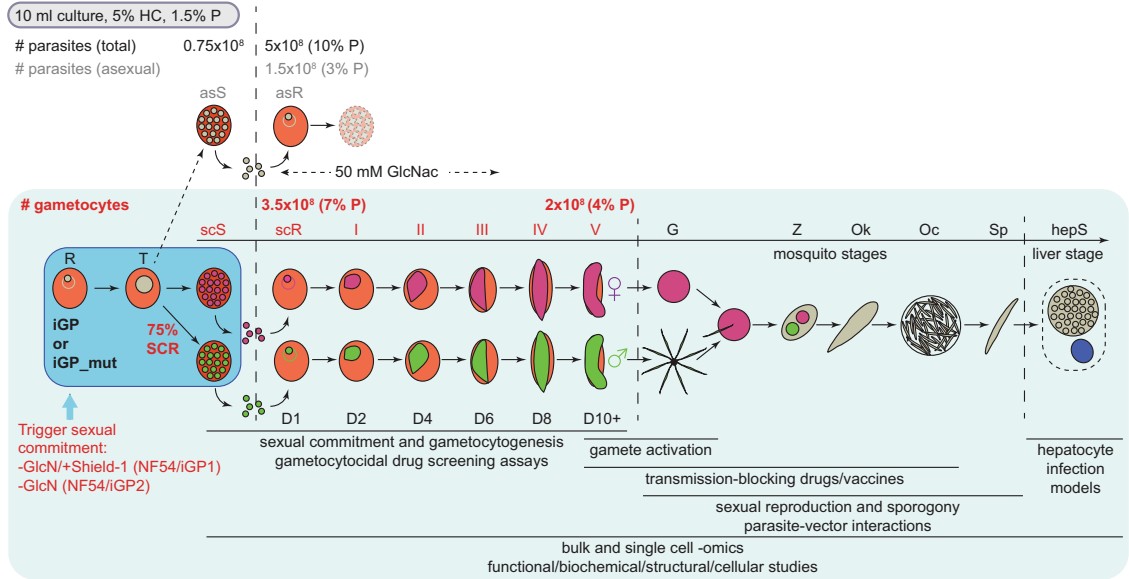

**Fig. 6 Scheme depicting the simple induction protocol for the routine mass production of NF54/iGP gametocytes and potential applications for future research.** Addition of Shield-1/removal of GlcN (NF54/iGP1) or removal of GlcN (NF54/iGP2) from a synchronous ring stage culture triggers sexual commitment in trophozoites and produces progeny consisting of up to 75% sexual ring stage parasites. Addition of 50 mM GlcNAc to the culture medium for the next six days eliminates the remaining asexual parasites. Expected numbers of parasite-infected RBCs (#) and percent parasitaemia (% P) of total (black letters), asexual (grey letters), and sexual (red letters) parasites in the progeny routinely obtained from a 10 ml culture at 5% haematocrit (HC) and 1.5% starting parasitaemia are indicated. Asexual parasites are depicted in grey, sexually committed parasites and gametocytes are depicted in purple (females) and green (males). asS/scS asexual/sexually committed schizont, asR/scR asexual/sexually committed ring stage, T trophozoite, I–V gametocyte stages I–V, D1–D10 days 1–10 of gametocyte maturation, G gametes, Z zygote, Ok ookinete, Oc oocyst, Sp sporozoites, hepS intrahepatic schizont. Possible applications of NF54/iGP lines for basic, applied, and translational research on *P. falciparum* gametocytes and mosquito-stage parasites are listed below the schematic.

In summary, we engineered marker-free inducible gametocyte producer lines that facilitate the routine mass production of synchronous gametocyte populations using a simple and reproducible experimental setup. We demonstrate that NF54/iGP1_D8 and NF54/iGP2_E9 parasites complete their life cycle in the mosquito vector and produce sporozoites that are infectious to human hepatocytes. We also showed that further genetic engineering of NF54/iGP parasites is straightforward. Hence, we believe the iGP approach developed in this study will become an invaluable and broadly applicable tool for fundamental, applied and translational research in the field of malaria-transmission biology.

## Methods

**Parasite culture.** *P. falciparum* 3D7 parasites were cultured using AB+ or 0+ human RBCs at 5% haematocrit and RPMI 1640 medium supplemented with 25 mM HEPES, 100 mM hypoxanthine, and 24 mM sodium bicarbonate and complemented with 0.5% Albumax II (Life Technologies). NF54 parasites were cultured with 10% AB+ human serum instead of 0.5% Albumax II. Growth medium was replaced daily. Intraerythrocytic growth synchronisation was achieved using repeated sorbitol treatments[93]. The NF54/iGP1, NF54/iGP2, and NF54/iGP2_NUP-mScarlet lines were cultured in the presence of 2.5 mM D-(+)-glucosamine hydrochloride (GlcN) to maintain *glmS* ribozyme activity during routine parasite propagation.

**Transfection constructs for CRISPR/Cas9-based gene editing.** Inducible *gdv1* transgene expression cassettes were inserted into the *cg6* (*glp3*) locus (PF3D7_0709200) of 3D7 and NF54 wt parasites by cotransfecting 50 μg each of a CRISPR/Cas9 transfection vector and a donor plasmid delivering the transgene cassette. 3D7/iGP1 was obtained by cotransfecting pHF_gC-*cg6* and pD_*cg6_cam-gdv1-gfp-dd* into 3D7 wt parasites. NF54/iGP1 was obtained by cotransfecting pBF_gC-*cg6* and pD_*cg6_cam-gdv1-gfp-dd-glmS* into NF54 wt parasites. NF54/iGP2 was obtained by cotransfecting pBF_gC-*cg6* and pD_*cg6_cam-gdv1-gfp-glmS* into NF54 wt parasites. NF54/iGP2_NUP313mSc was obtained by cotransfecting pBF_gC-*nup313* and pD_*nup313-mScarlet* into NF54/iGP2 parasites. All transfection plasmids cloned in this study are derivatives of the original pHF_gC/

pBF_gC CRISPR/Cas9 and pD donor plasmids published by Filarsky and colleagues[16] and are schematically displayed in Supplementary Figs. 1, 3 and 8.

The pHF_gC-*cg6* CRISPR/Cas9 plasmid was generated by T4 DNA ligase-dependent insertion of annealed complementary oligonucleotides (11 F, 11 R) encoding the single-guide RNA (sgRNA) target sequence sgt_*cg6* along with compatible single-stranded overhangs into *Bsa*I-digested pHF_gC[16]. The sgt_*cg6* target sequence (gcacaaatataaattaaatt) is positioned 24–44 bp downstream of the start codon of the *cg6* (*glp3*) gene and has been designed using CHOPCHOP[94]. pBF_gC-*cg6* was generated in the same manner using the pBF_gC vector[16].

The pD_*cg6_cam-gdv1-gfp-dd* donor plasmid was generated by Gibson assembly[95] of five PCR fragments encoding (1) the *P. falciparum* calmodulin (*cam*) promoter followed by a *gdv1-gfp-dd* fusion gene (amplified from pHcam-*gdv1-gfp-dd*[16] using primers 1 F and 1 R), (2) the *P. berghei* dihydrofolate reductase-thymidylate synthase (*pbdhfr-ts*) terminator sequence (amplified from the pH_gC vector[16] using primers 2 F and 2 R), (3) a 387 bp *cg6* 3′ homology region (HR) spanning bps 392–778 of the *cg6* coding sequence (amplified from 3D7 gDNA using primers 3 F and 3 R), (4) the plasmid backbone (amplified from pUC19 using primers 4 F and 4 R), and (5) a 276-bp *cg6* 5′ HR spanning bps −343 to −68 upstream of the *cg6* gene (amplified from 3D7 gDNA using primers 5 F and 5 R).

The pD_*cg6_cam-gdv1-gfp-dd-glmS* donor plasmid has been cloned by Gibson assembly using three fragments, namely (1) the *Eco*RI/*Age*I fragment of pD_*cg6_cam-gdv1-gfp-dd* encoding part of the *cg6* 3′ HR (bps 482–778 of the *cg6* coding sequence), the vector backbone, the *cg6* 5′ HR, the *cam* promoter, the *gdv1* gene, and bps 1–715 of the *gfp* coding sequence; (2) a PCR fragment encoding bps 699–714 of the *gfp* coding sequence followed by the *dd* and *glmS-246* sequence (amplified from pD_*ap2g-gfp-dd-glmS*[16] using primers 7 F and 7 R); (3) a PCR fragment encompassing the *pbdhfr-ts* terminator sequence and part of the *cg6* 3′ HR (spanning bps 392–501 of the *cg6* coding sequence) (amplified from pD_*cg6_cam-gdv1-gfp-dd* using primers 8 F and 8 R).

The pD_*cg6_cam-gdv1-gfp-glmS* vector was generated through Gibson assembly of two PCR fragments representing (1) the *pbdhfr-ts* terminator sequence, the *cg6* 3' HR, the vector backbone, the *cg6* 5′ HR, the *cam* promoter, the *gdv1* gene, and bps 1–714 of the *gfp* coding sequence followed by a TAA stop codon (amplified from pD_*cg6_cam-gdv1-gfp-dd-glmS* using primers 9 F and 9 R); and (2) the *glmS-246* sequence (amplified from pD_*ap2g-gfp-dd-glmS*[16] using primers 10 F and 10 R).

To tag the nucleoporin NUP313 (PF3D7_1446500) with mScarlet[96] in NF54/iGP2 parasites, the following CRISPR/Cas9 and donor plasmids were cloned. The pBF_gC-*nup313* vector was generated by T4 DNA ligase-dependent insertion of annealed complementary oligonucleotides (18 F, 18 R) encoding the sgRNA target sequence sgt-*nup313* along with compatible single-stranded overhangs into *Bsa*I-digested pBF_gC[16]. The sgt_*nup313* target sequence (gcactttgtagagataagta) is

positioned at 91–110 bp downstream of the *nup313* coding sequence. The pD_*nup313-mScarlet* donor vector is a result of Gibson assembly of five PCR fragments encompassing (1) a 941-bp *nup313* 5′ HR spanning bps 8063–9003 of the *nup313* coding sequence (amplified from NF54 gDNA using primers 13 F and 13 R), (2) the *mScarlet* gene (amplified from a *P. falciparum* codon-adjusted synthetic sequence (Genscript) (Supplementary Fig. 10) using primers 14 F and 14 R), (3) the *P. falciparum* histidine-rich protein 2 (*hrp2*) terminator sequence (amplified from pBF_gC with primers 15 F and 15 R), (4) a 1000 bp *nup313* 3′ HR spanning the region 90–1089 bp downstream of the *nup313* coding sequence (amplified from NF54 gDNA using primers 16 F and 16 R), and (5) the vector backbone amplified from pBF_gC using primers 17 F and 17 R. All oligonucleotide sequences used for cloning and Sanger sequencing are provided in Supplementary Table 1.

**Parasite transfection and selection of transgenic lines.** Parasites were transfected and gene-edited parasites selected as described by Filarsky and colleagues[16]. Briefly, RBCs derived from 5 ml of a synchronous ring stage culture (5–10% parasitaemia) were co-transfected with the matching pair of CRISPR/Cas9 and donor plasmids (50 μg each) using a Bio-Rad Gene Pulser Xcell Electroporation System (single pulse, 310 V, 250 μF). Twenty-four hours after transfection, 3D7/iGP parasites were selected with 5 nM WR99210 (WR) for the following six days and NF54/iGP1, NF54/iGP2, and NF54/iGP2_NUP313-mSc parasites were selected with 2.5 μg/ml blasticidin-S-HCl (BSD) for the following eleven days. Thereafter, all transfected parasites were cultured in the absence of drug pressure, until stably propagating populations were established. PCR on gDNA was performed to confirm proper genome editing and absence of plasmid DNA. The 3D7/iGP and NF54/iGP1 lines, which still retained the pHF_gC or pBF_gC CRISPR/Cas9 plasmids expressing the negative selection marker yFCU fused to hDHFR or BSD deaminase, respectively, were treated with 40 μM 5-fluorocytosine (5-FC) to obtain plasmid-free populations. 3D7/iGP, NF54/iGP1, and NF54/iGP2 lines were cloned out by limiting dilution using plaque assays[97]. All oligonucleotide sequences used for PCR on parasite gDNA and plasmid DNA are provided in Supplementary Table 1.

**Immunofluorescence assays.** α-GFP and α-Pfs16 IFAs were performed with methanol-fixed cells using mouse mAbs α-GFP (1:200) (Roche, #11814460001) and α-Pfs16 (1:500)[98] primary antibodies, and Alexa Fluor 488 goat α-mouse IgG secondary antibody (1:200) (Invitrogen, #A-11001). Staining of DNA and mounting of IFA slides was performed using Vectashield with DAPI (Vector Laboratories, #H-1200). Microscopy was performed using a Leica DM 5000B microscope with a Leica DFC 345 FX camera using a 63x immersion oil objective (total magnification = 1008x). All images were acquired via the Leica Application Suite software (version LAS 4.9.0) and processed using Adobe Photoshop CC with identical settings.

α-Pfg377 IFAs were performed with stage V gametocytes (day 10) fixed in methanol–acetone (60:40) using rabbit α-Pfg377 antibodies (1:1,000)[69] and Alexa Fluor 568 goat α-rabbit IgG (1:250) (Invitrogen, #A-11011). Nuclear DNA was stained during slide preparation with Vectashield containing DAPI (Vector Laboratories, #H-1200-10). Microscopy was performed using a Leica DM 5000B microscope with a Leica K5 camera using a 40x objective (total magnification = 400x). All images were acquired via the Leica Application Suite software (version LAS X) and processed using Fiji with identical settings.

IFAs on infected hepatocytes were performed on methanol-fixed cells with rabbit α-PfHsp70 (1:75) (StressMarq Biosciences, SPC-186), mouse α-Exp2 (1:1000) (The European Malaria Reagent Repository, 7.7), and chicken α-GFP (1:1000) (Invitrogen, #A10262) as primary antibodies and Alexa Fluor Plus 488 goat α-chicken IgY (Invitrogen, #A32931), Alexa Fluor 594 goat α-rabbit IgG (Invitrogen, #A32740), and Alexa Fluor 647 goat α-mouse IgG (Invitrogen, #A32728) secondary antibodies (all 1:400). All IFAs were performed in 96-well black/clear flat-bottom imaging microplates and DNA was stained with DAPI (Vector Laboratories, #H-1200). Confocal microscopy on infected hepatocytes was performed using a Zeiss LSM880 microscope with Airyscan using a 63x immersion oil objective (total magnification = 1008x). All images were acquired via the Zen Black software (version 14.0.18.201) and processed using Fiji with identical settings.

**Live-cell fluorescence microscopy.** To perform live-cell fluorescence imaging of NUP313-mScarlet presented in Fig. 4 and Supplementary Fig. 9, asexual blood stages were stained with 1 μg/mL DAPI (Biomol) in RPMI and gametocytes were stained with 4.5 μg/mL Hoechst33342 (Chemodex) in PBS at 37 °C for 15 min. Stained cells were pelleted at 1000 g for 1 min and the pellet resuspended in an equal volume of supernatant. About 10 μL of the sample were placed on a microscopy slide and covered with a cover slip. Fluorescence microscopy images were acquired with a Leica DM6 B microscope equipped with Leica DFC9000 GT camera using a 100x immersion oil objective (total magnification = 1000x). Filter block settings: mScarlet (Ex. 542–585 nm; Em. 604–644 nm), DAPI/Hoechst (Ex. 325–375; Em. 435–485). All images were acquired with the Leica Application Suite X software (LAS X) and processed using Adobe Photoshop CS2 with identical settings.

To perform live-cell fluorescence imaging of NUP313-mScarlet presented in Fig. 5, gametocytes were stained with 4.5 μg/mL Hoechst33342 (Sigma–Aldrich) in PBS at 37 °C for 20 min. Stained cells were pelleted at 300 g for 1 min and the pellet resuspended in an equal volume of the supernatant. About 3 μL of the sample were placed on a microscopy slide, mixed with 3 μL Vectashield, and covered with a cover slip. Images were acquired using a Leica DM 5000B microscope and a Leica DFC 345 FX camera using a 100x immersion oil objective (total magnification =1250x). Filter block settings: mScarlet (Ex. 543/22 nm; Em. 593/40 nm), Hoechst (Ex. 377/50 nm; Em. 447/60). All images were acquired using the Leica Application Suite software (LAS 4.9.0) and processed using ImageJ (version 1.52n) using identical settings.

**Induction of sexual commitment and gametocytogenesis.** Depending on the inducible GDV1 overexpression system employed, induction of sexual commitment was achieved by either stabilising GDV1-GFP-DD fusion protein expression via addition of Shield-1 (+Shield-1) (3D7/iGP), or stabilisation of *gdv1-gfp* mRNA via removal of GlcN (−GlcN) (NF54/iGP2 and NF54/iGP2_NUP313-mSc), or combining both treatments simultaneously (−GlcN/+Shield-1) (NF54/iGP1). To determine sexual conversion rates (SCRs), synchronous ring stage cultures at 1–3% parasitaemia (0–16 hpi, generation 1) were washed in culture medium and split into two identical cultures at 5% haematocrit. One was maintained under non-inducing conditions (control population representing background SCRs) and the other was induced for sexual conversion by triggering ectopic GDV1 expression through the addition of 1350, 675, or 337.5 nM Shield-1 (3D7/iGP), addition of 1350 nM Shield-1/removal of GlcN (NF54/iGP1), or removal of GlcN (NF54/iGP2, NF54/iGP2_NUP313-mSc). The inducing conditions were reversed 54 h later in the ring stage progeny (6–22 hpi, generation 2; day 1 of gametocytogenesis). To quantify SCRs, parasites were methanol-fixed at 36–44 hpi in generation 2 (day 2 of gametocytogenesis) and α-Pfs16 IFAs combined with DAPI staining were performed (see above). SCRs were determined as the proportion of early stage I gametocytes (DAPI-positive/Pfs16-positive) among all infected RBCs (DAPI-positive).

For gametocyte maturation assays, parasites were induced for sexual conversion as describe above. Culture medium containing 50 mM GlcNAc was added to the ring stage progeny (generation 2; day 1 of gametocytogenesis) and changed daily for six days to eliminate asexual parasites[53,68]. From day seven onward, gametocytes were cultured with normal culture medium. Gametocyte stages and gametocytaemia were assessed by visual inspection of Giemsa-stained thin blood smears prepared until day 12 of gametocytogenesis using a 100x immersion oil objective (total magnification = 1000x).

**Quantification of gametocyte sex ratios and exflagellation rates.** To determine the sex ratios of NF54/iGP1_D8 and NF54/iGP2_E9 gametocytes produced via induction of ectopic GDV1 expression, ring stage parasites were treated with −GlcN/+Shield-1 and −GlcN, respectively, and gametocytes cultured as explained in the section above. To obtain sufficient quantities of control gametocytes obtained without inducing ectopic GDV1 expression, sexual commitment rates of NF54/iGP1_D8 (+GlcN/−Shield-1) and NF54/iGP2_E9 parasites (+GlcN) were augmented using LysoPC-/choline-free minimal fatty acid medium (mFA) (RPMI 1640 medium, 25 mM HEPES, 24 mM sodium bicarbonate, 100 μM hypoxanthine, 0.39% fatty acid-free BSA (Sigma–Aldrich), 30 μM oleic acid, and 30 μM palmitic acid)[17]. To quantify female/male gametocyte sex ratios, IFAs using antibodies against the female-specific protein Pfg377[69] were performed to identify female (DAPI-positive/Pfg377-positive) and male stage V gametocytes (DAPI-positive/Pfg377-negative) as explained above.

To quantify male gametocyte exflagellation, 10 μL of stage V gametocyte cultures was added to 10 μL of 50 μM xanthurenic acid in incomplete RPMI 160 medium, placed into a Neubauer slide with rhodium-coated chamber bottom, and incubated for 10–15 min at room temperature. The exflagellation events were counted and the average per μl and 1% parasitaemia calculated.

**Western blot analysis.** Synchronous ring stage parasites were split at 0–8 hpi and treated with +GlcN/−Shield-1, −GlcN/−Shield-1 or −GlcN/+Shield-1 (NF54/iGP1_D8), and +GlcN or −GlcN (NF54/iGP2_E9). At 34–42 hpi, schizonts were released from the iRBC by saponin lysis (0.15% in PBS) and whole-cell extracts prepared by suspension of the parasite pellet in UREA/SDS lysis buffer [(8 M urea, 5% SDS, 50 mM Bis-Tris, 2 mM EDTA, pH 6.5, 1 mM TCEP, and 1x protease inhibitor (Roche)]. To prepare protein extracts from gametocytes, parasite cultures were induced for ectopic GDV1 expression by treatment with −GlcN/+Shield-1 (NF54/iGP1_D8) or −GlcN (NF54/iGP2_E9). The ring stage progeny were split (generation 2; day 1 of gametocytogenesis) and cultured separately under +GlcN/−Shield-1 or −GlcN/−Shield-1 conditions (NF54/iGP1_D8) and under +GlcN or −GlcN conditions (NF54/iGP2_E9) with 50 mM GlcNAc added to the culture medium for six days to eliminate asexual parasites[53,68]. Whole-cell extracts were prepared from stage III gametocytes (day 6) as described above. Protein extracts were separated on a NuPage 4–12% Bis-Tris gel (Novex) using NuPage MES SDS Running Buffer (Novex). Proteins were detected with primary antibodies mouse mAb α-GFP (1:1,000) (Roche Diagnostics #11814460001) and rabbit α-PfHP1 (1:14,000)[12], and secondary antibodies α-mouse IgG (H&L)-HRP (1:10,000)

(GE Healthcare #NXA931) and α-rabbit IgG (H&L)-HRP (1:10,000) (GE Healthcare #NA934). Chemiluminescence signals were detected using the KPL LumiGLO Reserve Chemiluminescent Substrate Kit (SeraCare #5430-0049).

**Standard membrane feeding assays (SMFAs)**. On day 10, 13, and 14 of gametocytogenesis, aliquots of stage-V gametocyte cultures were tested in the Standard Membrane Feeding Assay (SMFA)[65] according to an established workflow[70]. Briefly, after a final change of culture medium on the day of mosquito feeding, a small volume of the gametocyte culture was used to prepare the blood meal and to make blood smears. About 300 μL of the gametocyte culture (approximately 2.5% hamatocrit and 3–10% gametocytemia) was mixed with prewarmed 180 μL of packed RBCs and the cells were pelleted for 20 sec. The supernatant was carefully removed and the RBC pellet resuspended in 150 μL of prewarmed human serum. Small membrane feeders were used to feed female *Anopheles stephensi* mosquitoes separately with each of the prepared gametocyte populations (approximately 0.15–0.5% gametocytemia and 50% hematocrit). For sporozoite production experiments, midi membrane feeders were used with three times the above volumes with the ratios consistent to feed 100 mosquitoes[65]. On day 8 after feeding, midguts of 20 mosquitoes per feed were dissected and stained for oocysts using 3% mercurochrome. The numbers of oocysts per midgut were counted by microscopy (1000x magnification). For sporozoite collection on day 17, at least 25 mosquitoes were dissected with salivary glands collected and crushed in complete Williams B media at room temperature. Sporozoites were counted using a Neubauer slide with rhodium-coated chamber bottom and used directly for hepatocyte infection assays.

**Hepatocyte infection assay**. Fresh human hepatocytes were isolated from patients undergoing a partial hepatectomy and seeded ($6 \times 10^4$ hepatocytes/well) onto collagen-coated 96-well black/clear flat-bottom TC-treated imaging microplates (Corning, #353219) in complete Williams B medium and incubated at 37 °C in 5% $CO_2$[99]. On day 3 after seeding, sporozoites were added ($6 \times 10^4$ sporozoites/well), spun down at 500 g for 5 min, and allowed to invade hepatocytes for three hours. The wells were washed in complete Williams B medium and incubated at 37 °C in 5% $CO_2$, with the medium changed every day[99]. Plates were fixed with ice-cold methanol on days 3 and 5 and used for IFA analysis as described above.

**Ethical statement**. Primary human liver cells were freshly isolated from remnant surgical material. The samples were anonymized, and general approval for use of remnant surgical material was granted in accordance with the Dutch ethical legislation as described in the Medical Research (Human Subjects) Act and confirmed by the Committee on Research involving Human Subjects, in the region of Arnhem–Nijmegen, the Netherlands.

**Statistics and reproducibility**. All data from assays quantifying sexual commitment rates, gametocyte staging, gametocyte sex ratios, exflagellation rates, mosquito infection rates, and oocyst and sporozoite production are represented in graphs showing individual data points and the mean with error bars defining the standard deviation, or the median and interquartile ranges as indicated in the figure legends and the Source data file. To test for statistical significance of the difference in gametocyte sex ratios, a paired two-tailed Student's *t* test was used as indicated in the figure legend. The exact number of biological replicates performed per experiment and the number of cells analysed per sample are indicated in the figure legends and the Source data file. Data were analysed using Excel 2016 and plotted using GraphPad Prism (version 8.2.1).

**Reporting summary**. Further information on research design is available in the Nature Research Reporting Summary linked to this article.

## Data availability
All data generated or analysed during this study are included in this published article and its associated Supplementary Information and Source data files. The associated Source data file contains the raw data underlying the graphs presented in Figs. 1d, 2b, 2e, 2g, 3a, 3b and Supplementary Figs. 2b, 5a, 6b–d, 7a-b. Correspondence and requests for materials should be addressed to T.S.V. Source data are provided with this paper.

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

## Acknowledgements

This work was supported by the Swiss National Science Foundation (grant numbers BSCGI0_157729 and 31003A_169347) and the Fondation Pasteur Suisse. A.A. received a scholarship from the Jürgen Manchot Foundation. A.S., M.V.B., G.G., R.W.S., and N.I.P. are supported by the European Union's Horizon 2020 research and innovation programme under grant agreement No. 733273. T.W.A.K. is supported by the Netherlands Organisation for Scientific Research (NWO-VIDI 864.13.009).

## Author contributions

S.D.B. cloned all transfection plasmids, generated and characterised all transgenic cell lines, and designed, performed, and analysed experiments related to the NF54/iGP lines. A.P. designed, performed, and analysed experiments related to the 3D7/iGP line. A.A and N.M.B.B performed fluorescence microscopy on the NF54/iGP2_NUP313-mSc line. E.C. performed Western blots and assays to quantify sexual conversion rates and gametocyte sex ratios, M.V.B. and G.G. performed and analysed mosquito infection experiments. A.S. and N.I.P performed and analysed hepatocyte infection experiments. T.W.G, H.P.B, R.W.S., N.I.P., and T.W.A.K. provided conceptual advice and resources and supervised experiments. S.D.B., A.P., A.A., N.M.B.B., and N.I.P. helped preparing illustrations. T.S.V. conceived the study, designed, supervised, and analysed experiments, provided resources, prepared illustrations, and wrote the paper. All authors contributed to editing of the paper.

## Competing interests

The authors declare no competing interests.
