## [Peer Review File · Nature Communications]

Reviewer comments, first round –

REVIEWER COMMENTS

Reviewer #1 (Remarks to the Author):

The manuscript describes a new method based on genome editing that significantly increases the sexual commitment rate of *P. falciparum*. The inducible expression of a regulator gene called GDV1 achieves conversion rate of about 75%, which represents a significant improvement over traditional methods. Importantly, the authors show that the produced gametocytes remain highly infective for *Anopheles* mosquitoes based on oocyst counts. Furthermore, the marker-free NF54/iGP line can be further used to genetically manipulate genes of choice for more specific questions linked to sexual stage development. These new parasite lines represent a very valuable tool for the community. The manuscript is clearly written with a very detailed Introduction section. This section could be shortened in case space is an issue.

Main points :

1. Conditional activation has been previously reported for *P. berghei* and very recently for *P. falciparum* by genome editing (DiCre-mediated promoter flipping of the *ap2-g* gene) achieving up to 90% conversion rates. Those publications might be considered as an issue for publication of this work. However, the published systems use a selectable marker for the genome editing step whereas this manuscript has the merit to have validated better their inducible marker-free system for studies in mosquitoes. In addition, this work shows that the pathway of GDV1 induction differs from the direct activation of *ap2-g* published earlier.

Minor points :

2. What is the level of GDV1-GFP-DD-glmS in the repressed state compared to levels in wild type parasites? Western blot data would be needed to demonstrate that the system is not leaky. IFA is used to address this question but a quantitative measure would be more appropriate here.

3. Has the mosquito infection of the NF54/iGP clones analysed for sporozoites production in salivary glands/sporozoite infection of hepatocytes. This would validate further the new system for studies to those important research fields. Please comment.

Reviewer #2 (Remarks to the Author):

Overall thoughts and summary:

This is a very timely manuscript which presents yet another alternative for generating large numbers of *Plasmodium* gametocytes. This was first demonstrated in the *P. berghei* rodent malaria parasites and then in *P. falciparum* using a different strategy. However, in this manuscript, Boltryk et al. have elegantly put forth a new strategy for generating parasite lines to be used as future tools. I fully agree with the authors that a universal strategy, such as the one proposed in this manuscript, is critical if we want to significantly advance our knowledge of gametocyte biology. The authors investigate the ability of GDV1-overexpressing *Plasmodium falciparum* parasites to produce vastly increased numbers of the sexual stages (gametocytes) of the parasite's life cycle. Previous results have shown the expression of the master regulator of *P. falciparum* gametocytogenesis, AP2-G results in the massive production of mature gametocytes. The results in this manuscript expand on this previous work by showing that the GDV1-overexpressing parasites are also capable of completing the parasite life cycle through the mosquito stages and provide a significant advance in marker-free parasite lines for further modification. However, this manuscript presents two options for generating gametocytes: one requiring the addition of Shld1 and removing GlcN or, alternatively, simply removing Shld1.

I think the authors should decide which is the better option for a standard protocol for the field and state why? Once they decide this, why not frame the manuscript around presenting this as the way forward for the field for the study of gametocytes?

Also, the quantification of the phenotype of the generated lines is incomplete and a major omission is the lack of any comparison to a wild type gametocyte producing line. Ideally, a direct comparison between the conversion rates of this new parasite line and the AP2-G overexpression line would have been even more convincing. Here, the manuscript suffers from a lack of further analysis of the molecular contents of these parasites, i.e. the transcriptome as was reported for the AP2-G overexpression line. Generally, the parasite lines should be described in more detail given that they represent the central result of the paper.

I might recommend a slightly different approach to framing this paper. Why not state that the objective of this work is to create a parasite line with maximal conversion to gametocytes that retains all of the properties required for transmission to mosquitoes. State that the optimal strain for this is currently NF54 and then describe the development of an exciting new tool for the field that will open up many possibilities for the field, including among them the ability to better screen for transmission blocking drugs, which is a noteworthy goal and achievement for the field.

Major Comments:

Surprisingly, the bulk of the manuscript is constituted by an exceptional Introduction and a detailed Discussion. The Introduction which goes on for four pages is an excellent review of the current understanding of gametocyte commitment, conversion, and development, but for the purposes of a research article it is far too long. The Discussion similarly goes on at length extolling the many possible virtues of the developments presented in this manuscript, but again this would be best saved for a review. In the end, it distracts from the message of the manuscript. The result is that it leaves the Results section as the least impactful in the end.

My main concern with this work is that the authors do not explore the possible side effects that may arise due to the over-expression of GDV1. Although it is clear that epigenetic regulation at the *pfap2-g* locus must be affected to result in high gametocyte production, any/all loci that are regulated via HP1 will also be affected. Therefore, I am interested in knowing what other mechanisms are perturbed and would these not potentially impact the natural development of gametocytes which would generally not express all of these genes that are normally repressed via HP1/H3K9me3 regulatory action.

I don't understand why the section on 3D7 is included as an entire section of the results. Perhaps as Supplementary data, but clearly these lines are inferior to using the NF54.

The authors introduce a GDV1+tag expression cassette into the *cg6* locus in both the 3D7 and NF54 parasite background lines. They explain that the chosen locus is dispensable, but what is the purpose of using this locus over the GDV1 endogenous locus? And what are the consequence of leaving a functional endogenous copy of *gdv1* in the genome? Is there any possibility that the wild-type locus is having an effect on this phenotype?

From the description in the methods it seemed like the authors were culturing on 2.5 mM glucosamine long-term, how does this affect the growth of the parasites in vitro compared to a wild type parasite line? It is known that certain concentrations of glucosamine are toxic in vitro and it is necessary for these effects to be quantified. In addition, the effect of addition of N-acetyl glucosamine to deplete asexual parasites during gametocyte development of *glmS*-containing parasites is unknown.

The authors do not show that they have quantified the level of perturbation of GDV1 expression achieved with any of the modifications that were made to the locus in the NF54 lines. Evidence of induction is lacking and would be nice, although the phenotype is clear. How much more protein is in the cell? Minimally, a Western Blot that shows restoration of protein expression following Shield-1 addition or ablation of protein expression following glucosamine addition is required. In addition, due to off target cutting by Cas9, it would be beneficial to include whole genome sequencing of the

modified parasite lines to verify that there are no off-target modifications. This is now standard practice.

In general, what happens to these parasite lines in long-term culture? From the early work of Eksi et al. it is known that the *gdv1* locus can be lost during in vitro culture. Is it possible that the incorporation of a 2nd non-selectable copy of *gdv1* (in the *cg6* locus) could also be selected against in culture? Have the authors explored this idea?

The authors use two different methods of knockdown in the NF54 background line. The first attempt using just the FKBPdd-mediated knockdown did not produce viable transfectants. So, the authors continued on to use a double knockdown method (FKBPdd+glmS) and an alternative single knockdown method (glmS). Both methods knocked-down GDV1, yet at two different levels: the protein level for FKBPdd and the RNA level for glmS. Why continue to use FKBPdd in the double knockdown system if by itself it would not produce a viable transfectant? Why not just move forward with the glmS knockdown method? Their results show that both methods could not produce more than 75% conversion, so which method would be the best option for someone wanting to use this method?

The authors speculate on Page 8 that they were unable to integrate the GDV1-GFP-DD into the *cg6* locus due to a deficiency in DD-dependent degradation in the NF54 strain. If this is true, can they demonstrate this with the GDV1-GFP-DD-glmS line that is successfully integrated into the *cg6* locus? In the absence of both glucosamine and *Shld-1*, the authors should be able to test whether the strain 1) stabilizes GDV1 and 2) produces more gametocytes than the parental NF54 merely due to the presence of residual GDV1 that is inefficiently degraded.

In this manuscript two NF54 lines are presented, both of which can transmit to mosquitoes. In my opinion, for this to truly become relevant to the malaria community at large, the authors should present a compelling reason to use one over the other, either the GDV1-GFP-DD-GlmS or the GDV1-GFP-GlmS.

Although the authors state in the discussion that they do not know why they fail to get 100% conversion, this should be explored further. Again, is this perhaps due to the presence of the endogenous *Gdv1* protein?

Figure Comments:

Figure 1b is not necessary as a main panel and can be moved to the Supplementary data.

Figure 2: The description of the conversion rate assay is difficult to interpret. It is not clear how these conversion rates are quantified? Are these rates a function of gametocytemia/ rings or a percentage of the total culture on day 2?

Pfs16 is also expressed in asexual stages and the time of sampling was too early in development for the line to be enriched for gametocytes. From the IFA it would seem that the authors are counting schizonts that are expressing *Pfs16* as gametocytes, however, there are no data showing all *Pfs16*-expressing asexual parasites will develop into gametocytes, which would result in an overestimation of gametocytemia.

The experiment would be much easier to interpret with a WT NF54 control to relate back to. Without knowing the basal commitment rate it's difficult to relate the scope of improvement in the transgenic line.

Why are the conversion rates in the clonal lines lower? Were the experiments in the clonal line replicated? According to the figure legend, I think not, but if so, please show the data. If not, please provide additional replicates. Repetition would show whether it is a statistical oddity or a real biological effect? There are often big variations in commitment between inductions, it is not convincing enough to have a single experiment as a defining result for the paper. Since the non-clonal lines contain parasites with the episomal plasmids/ concatamerized plasmids, is the hypothesis that these parasites result in better conversion? It is odd that a mixed culture would produce a better result reproducibly?

Does the variability in conversion between strains in Figures 2b and 2d arise from differences in expression of Gdv1 from the endogenous *gdv1* locus? The authors should test this.

Why is the mother line tested in Figure 2b and 2d? What is the relevance? And in the figure legend the term "progeny of the NF54/iGP1 mother line" should either be "progeny clones" or just "clones".

In Figure 3, it appears that the DD-containing parasite is less effective at producing oocysts at all feeding days. Does this imply perhaps that there may be a secondary role for GDV1 that is not able to function in the absence of Shld-1? While the strain lacking the DD (only regulated by the *glmS*) continues to express the protein, allowing for proper function in many?, some?, all? cells.

Also Figure 3: It would be vastly preferred if the authors had performed their own NF54 mosquito infection control with the strain used to transfect the plasmids into if possible.

For Figure 3 were the same number of cells used? The methods say approximately 3%, but were they set up to match as closely as possible?

Figure 4a can be reduced in size. Perhaps the microscopy can be side-by-side rather than stacked on top for asexual and gametocyte stages?

Figure 4&5: Tagging a nucleoporin to characterize the nuclear boundaries of the gametocyte is a nice addition as proof-of principle for making additional modifications in this line. However, without doing the same in a control strain, that does not modify a gene involved in transcriptional regulation within gametocytes, it is difficult to conclude from these data that the nuclear structure of the gametocyte is not modified by the overexpression of GDV1.

Figure 6 is a nice overview, but there is nothing in this work demonstrating that oocysts can go on to produce viable sporozoites that can infect hepatocytes. If such data exist, please add them as they would strengthen the manuscript. Otherwise, consider shortening the figure at the oocyst stage.

Minor Comments:

Page 6: Change "...non-proliferative cells and thus rapidly overgrown..." to "...non-proliferative cells and thus are rapidly overgrown..."

Page 8: Change "...the sexual ring stage progeny was exposed to 50 nM ... to eliminate asexual parasites and was thereafter..." to "...the sexual ring stage progeny were exposed to 50 nM ... to eliminate asexual parasites and were thereafter..."

Page 11: should be nuclear lamina, currently "lamin"

Page 11: Reference formatted incorrectly (Field and Rout, 2019).

Page 13: Discussion: "Fourth, gametocyte synchronicity and yield are highly reproducible between experiments; starting with approximately 0.75×10^7 iGP ring stage parasites in the commitment cycle (5% hematocrit, 1.5% parasitemia) routinely delivers up to 2×10^8 stage V gametocytes per 10 ml of culture volume in a single induction experiment..." The data that led up to this observation would be much better served with quantification in the manuscript than merely described in the discussion.

Reviewer #3 (Remarks to the Author):

The development of gametocytes during the blood stage of the malaria parasite's life cycle is crucial for malaria transmission. It has been difficult to study the process of gametocytogenesis because not all strains produce gametocytes and the conditions to trigger gametocytogenesis are not understood. However, significant progress has been made in our understanding of the genetic

regulation of gametocytogenesis in malaria parasites in recent years. The authors have previously made observations that the nuclear protein, gametocyte development 1 (GDV1), is essential for gametocytogenesis in *P. falciparum*. In this manuscript, the authors have used this observation to develop a method to develop a transgenic *P. falciparum* line in which expression of GDV can be switched on in blood stage cultures to trigger the production of gametocytes. This approach provides a reliable and efficient production of synchronized *P. falciparum* gametocytes providing a useful tool to produce gametocytes and study the process of gametocytogenesis. A similar approach in which PfAP2-G was conditionally expressed in a transgenic *P. falciparum* line to produce gametocytes has been previously described (Llora-Battle et al., *Sc Adv* 2020). However, male gametocytes did not exflagellate and therefore mosquito transmission was not achieved. Here, although the authors do not specifically analyse formation of male and female gametocytes and the competence of latter to exflagellate, they do demonstrate the ability of induced gametocytes to infect mosquitoes and form oocysts.

The authors should address the following comments:

1. The authors do not actually demonstrate changes in levels of GDV1 in presence/absence of Shield 1 and GlcN as appropriate in the different transgenic lines. The authors should perform Western blot analysis to demonstrate changes in GDV1 levels following addition/removal of Shield 1 and GlcN.
2. Is there any difference in male/female gametocyte ratios following induction of gametocytogenesis by expression of GDV1? Is the efficiency of exflagellation of male gametocytes following induction of gametocytes by expression of GDV1 in transgenic line similar to that in wild type strain? Similarity in such phenotypes will confirm that this model does in fact reflect the normal process of gametocytogenesis in wild type parasites.
3. The authors further genetically manipulate the transgenic gametocyte producing lines to tag a nucleoporin (PfNUC313) with a 'scarlet' fluorescent protein to follow the distribution of nuclear pores during the process of gametocytogenesis in gametocytes at different stages. While they are successful at doing so, these data are purely descriptive. It is not clear what question or hypothesis these experiments were trying to address.

Author's response letter

We thank the three reviewers for their critical assessment and important input on our manuscript.

Reviewer #1 (Remarks to the Author):

The manuscript describes a new method based on genome editing that significantly increases the sexual commitment rate of *P. falciparum*. The inducible expression of a regulator gene called GDV1 achieves conversion rate of about 75%, which represents a significant improvement over traditional methods. Importantly, the authors show that the produced gametocytes remain highly infective for *Anopheles* mosquitoes based on oocyst counts. Furthermore, the marker-free NF54/iGP line can be further used to genetically manipulate genes of choice for more specific questions linked to sexual stage development. These new parasite lines represent a very valuable tool for the community. The manuscript is clearly written with a very detailed Introduction section. This section could be shortened in case space is an issue.

>>>We thank reviewer 1 for her/his appreciation of our work. We shortened the Introduction by half a page. We still kept it comprehensive though as we strongly feel the broad readership of *Nature Communications* will profit from a detailed introduction of the topic and its relevance.

Main points:

1. Conditional activation has been previously reported for *P. berghei* and very recently for *P. falciparum* by genome editing (DiCre-mediated promoter flipping of the ap2-g gene) achieving up to 90% conversion rates. Those publications might be considered as an issue for publication of this work. However, the published systems use a selectable marker for the genome editing step whereas this manuscript has the merit to have validated better their inducible marker-free system for studies in mosquitoes. In addition, this work shows that the pathway of GDV1 induction differs from the direct activation of ap2-g published earlier.

>>>It is true that the studies employing DiCre-mediated ap2-g promoter flipping approaches published by the Waters lab (*P. berghei*) and Cortes lab (*P. falciparum*) achieve higher sexual conversion rates (SCRs). Both studies employed this system primarily to generate new insight into the processes linked to sexual commitment and early gametocyte differentiation. Our study follows a very different aim. We wished to exploit the native epigenetic mechanisms underlying ap2-g activation to develop a reliable and broadly applicable biological tool for basic and applied research in the field of *P. falciparum* malaria transmission. We successfully accomplished this by engineering clean marker-free cell lines and clones carrying a single inducible ectopic GDV1 transgene cassette, leaving the ap2-g locus unmodified and subject to intrinsic control mechanisms. We demonstrate that our iGP parasites produce massive amounts of synchronous and fully viable gametocytes (75% SCRs), and these gametocytes are infectious to mosquitoes (initial data and new data from additional SMFA experiments; Figs. 3 and S7). Importantly, we now also show that the NF54/iGP lines produce viable sporozoites able to infect hepatocytes (new data; Figs. 3 and S7). While the ap2-g promoter switch line from the Cortes lab achieves higher SCRs (90%), these gametocytes carry an irreversibly modified ap2-g locus and are non-infectious to mosquitoes, and hence don't offer the many opportunities for future research on late stage gametocytes and mosquito stage parasites that our iGP lines offer. We therefore don't think the existing studies are an issue for publication of our work.

Minor points:

2. What is the level of GDV1-GFP-DD-glmS in the repressed state compared to levels in wt parasites? Western blot data would be needed to demonstrate that the system is not leaky. IFA is used to address this question but a quantitative measure would be more appropriate here.

>>>We are unable to compare expression levels of endogenous GDV1 and ectopic GDV1-GFP(-DD) since we lack an antibody against GDV1. We now performed two independent

Western blot experiments to detect GDV1-GFP(-DD) expression levels in NF54/iGP1_D8 and NF54/iGP2_E9 schizonts under non-inducing and inducing conditions. The Western blot results show that GDV1-GFP(-DD) expression is only detectable under culture conditions that induce ectopic GDV1 expression (new data; Figs. 2f and S5b). The functional readout obtained from triplicate SCR assays performed in parallel are fully consistent with the Western blot data (new data; Fig. 2e). The background SCRs under non-inducing conditions were low for both clones (8-9%), but higher compared to NF54 wt parasites (2%). Whether these slightly increased SCRs are due to leaky expression of ectopic GDV1 or to unrelated inter-clonal/inter-strain variation is unknown (we mention this in the Results section). However, if leaky expression indeed occurs, it occurs at very low levels undetectable by our Western blots and is functionally irrelevant as the background SCRs are low. We believe this provides sufficient proof to show that the GDV1-GFP-DD-glmS and GDV1-GFP-glmS systems are robust, and that NF54/iGP feeder cultures can easily be propagated under non-inducing conditions and instantly be induced for gametocyte mass production at any time.

3. Has the mosquito infection of the NF54/iGP clones analysed for sporozoites production in salivary glands/sporozyte infection of hepatocytes. This would validate further the new system for studies to those important research fields. Please comment.

>>>As mentioned above, we now performed new SMFA and hepatocyte infection experiments. We show that the NF54/iGP1_D8 and NF54/iGP2_E9 clonal lines infect mosquitoes and produce oocysts and salivary gland sporozoites as efficiently as NF54 wt gametocytes, and that these sporozoites are able to infect and mature within hepatocytes (new data; Figs. 3 and S7).

Reviewer #2 (Remarks to the Author):

Overall thoughts and summary:

This is a very timely manuscript which presents yet another alternative for generating large numbers of Plasmodium gametocytes. This was first demonstrated in the P. berghei rodent malaria parasites and then in P. falciparum using a different strategy. However, in this manuscript, Boltryk et al. have elegantly put forth a new strategy for generating parasite lines to be used as future tools. I fully agree with the authors that a universal strategy, such as the one proposed in this manuscript, is critical if we want to significantly advance our knowledge of gametocyte biology. The authors investigate the ability of GDV1-overexpressing Plasmodium falciparum parasites to produce vastly increased numbers of the sexual stages (gametocytes) of the parasite's life cycle. Previous results have shown the expression of the master regulator of P. falciparum gametocytogenesis, AP2-G results in the massive production of mature gametocytes. The results in this manuscript expand on this previous work by showing that the GDV1-overexpressing parasites are also capable of completing the parasite life cycle through the mosquito stages and provide a significant advance in marker-free parasite lines for further modification. However, this manuscript presents two options for generating gametocytes: one requiring the addition of Shld1 and removing GlcN or, alternatively, simply removing Shld1.

>>>We thank reviewer 2 for her/his appreciation of our work.

1. I think the authors should decide which is the better option for a standard protocol for the field and state why? Once they decide this, why not frame the manuscript around presenting this as the way forward for the field for the study of gametocytes?

>>>We engineered the NF54/iGP1 (DD-glmS) and NF54/iGP2 (glmS only) lines since we were unable to transfer the DD-only system (that worked so beautifully in 3D7/iGP parasites) to NF54 parasites. To be on the safe side we decided to add the glmS element to the existing GDV1-GFP-DD module for NF54/iGP1. As we didn't know if this double conditional system would lead to high sexual induction rates, we also tested the glmS element alone as an alternative (NF54/iGP2). We were very satisfied that both approaches were successful and that both lines, and clones thereof, achieved very high sexual conversion rates (SCRs).

At this stage, however, we couldn't just assume that both lines would produce exflagellation-competent gametocytes that are infectious to mosquitoes and produce infectious sporozoites. We therefore had to take both lines and both clones all the way through from sexual conversion assays to sporozoite production, and we did this very rigorously to obtain solid data. In this revised version we placed our focus on the two clonal lines as we think that in order to serve as broadly applicable tools well characterised clonal lines are superior to uncloned populations.

Based on our results, both the NF54/iGP1_D8 and NF54/iGP2_E9 clones have excellent properties and are equally suitable for future research. Depending on the specific research question or application, one clone may be slightly more attractive than the other. We therefore prepared Table S2 in which we compare the properties of each clone and give personal recommendations for their use for future research.

2. Also, the quantification of the phenotype of the generated lines is incomplete and a major omission is the lack of any comparison to a wt gametocyte producing line. Ideally, a direct comparison between the conversion rates of this new parasite line and the AP2-G overexpression line would have been even more convincing. Here, the manuscript suffers from a lack of further analysis of the molecular contents of these parasites, i.e. the transcriptome as was reported for the AP2-G overexpression line. Generally, the parasite lines should be described in more detail given that they represent the central result of the paper.

>>>SCRs of wt parasites are low (<10%) and this has been reported in numerous publications (as mentioned in our Introduction section). We now still performed triplicate experiments with NF54 wt parasites and obtained SCRs of 2% (new data; Fig. 2e). In absence of GDV1-GFP-(DD) induction, our NF54/iGP lines and clones show default SCRs of 5-9% (Fig. 2b and 2e), which is slightly higher but still typical for NF54 wt parasites and demonstrates efficient prevention of GDV1-GFP-(DD) expression by the DD-glms (NF54/iGP1) or glmS only (NF54/iGP2) systems. In addition, we now performed exflagellation assays (new data; Figs. S6c and d), additional SMFA experiments (new data; Figs. 3a, 3b, S7a and S7b) and hepatocyte infection experiments (new data; Figs. 3c and S7c) with the NF54/iGP lines and clones and with NF54 wt parasites. These combined results show that the NF54/iGP1_D8 and NF54/iGP2_E9 clones exflagellate, infect mosquitoes, produce oocysts and infectious sporozoites as efficiently as NF54 wt parasites. We already highlighted and discussed in our manuscript the 90% SCRs that are consistently achieved by the ap2-g promoter switch line from the Cortes lab. As these results are well documented in the Cortes paper (Llora-Batlle et al., Sci Adv 2020) we can't understand why these experiments should be repeated again and compared in our laboratory.

With regard to the further analysis of the "molecular contents" of the NF54/iGP lines, we performed a series of additional experiments. We quantified GDV1-GFP-(DD) expression levels by Western blot in schizonts (new data; Figs. 2f and S5b) and in gametocytes (new data; Fig. S5c). The results show that ectopic GDV1 is not expressed in NF54/iGP1 gametocytes. Low level expression is detectable in NF54/iGP2 gametocytes but this can be prevented by addition of GlcN to the culture medium. We also didn't observe ectopic GDV1 expression in oocysts/sporozoites by live cell fluorescence microscopy or in liver stages by IFA (new data; Figs. 3 and S7). We also show that induction of gametocytogenesis via ectopic GDV1 expression does not alter female/male gametocyte sex ratios (new data; Figs. 2g, S6a and S6). As already mentioned above, we also demonstrate that exflagellation capacity, mosquito infectivity (oocyst and sporozoite prevalence and intensity) and hepatocyte infection of NF54/iGP1_D8 and NF54/iGP2_E9 are not impaired compared to NF54 wt parasites (new data; Figs. 3, S6c, S6d and S7)

The transcriptomic changes in response to GDV1 overexpression have already been reported in detail in our previous study (Filarsky et al., Science 2018). These results showed that GDV1 overexpression triggers the natural AP2-G-dependent transcriptional cascade of sexual conversion and early gametocyte development and has no effect on the transcription of other genes. In other words, overexpression of GDV1 triggers the same specific transcriptional programme that is also induced by AP2-G overexpression (Kafsack et al.,

Nature 2014; Llorca-Batlle et al., Sci Adv 2020). This is entirely expected as GDV1 is a specific and required activator of AP2-G expression (Eksi et al., PLoS Pathogens 2012; Filarsky et al., Science 2018; Usui et al., Nat Commun 2019).

3. I might recommend a slightly different approach to framing this paper. Why not state that the objective of this work is to create a parasite line with maximal conversion to gametocytes that retains all of the properties required for transmission to mosquitoes. State that the optimal strain for this is currently NF54 and then describe the development of an exciting new tool for the field that will open up many possibilities for the field, including among them the ability to better screen for transmission blocking drugs, which is a noteworthy goal and achievement for the field

>>>We thank reviewer 2 for this comment. We incorporated this recommendation accordingly in the Introduction section.

Major Comments:

4. Surprisingly, the bulk of the manuscript is constituted by an exceptional Introduction and a detailed Discussion. The Introduction which goes on for four pages is an excellent review of the current understanding of gametocyte commitment, conversion, and development, but for the purposes of a research article it is far too long. The Discussion similarly goes on at length extolling the many possible virtues of the developments presented in this manuscript, but again this would be best saved for a review. In the end, it distracts from the message of the manuscript. The result is that it leaves the Results section as the least impactful in the end.

>>>We deliberately wrote extensive Introduction and Discussion sections to provide the broad readership of Nature Communications with comprehensive background information on the complex and relevant topic of gametocyte biology and malaria transmission, and to highlight and discuss the importance and implications of our work. We similarly believe it is important to share our thoughts about the many possibilities offered by the NF54/iGP lines for future research. In the revised manuscript, both sections are still extensive. However, we restructured and shortened the Introduction section by half a page. The revised Discussion section is also slightly shorter despite the increased focus on the two NF54/iGP clones and the many new results included. We are willing to shorten these sections further should this become a condition for publication.

5. My main concern with this work is that the authors do not explore the possible side effects that may arise due to the over-expression of GDV1. Although it is clear that epigenetic regulation at the *pfap2-g* locus must be affected to result in high gametocyte production, any/all loci that are regulated via HP1 will also be affected. Therefore, I am interested in knowing what other mechanisms are perturbed and would these not potentially impact the natural development of gametocytes which would generally not express all of these genes that are normally repressed via HP1/H3K9me3 regulatory action.

>>>We would like to point out that the consequences of GDV1 overexpression have been described in detail at the transcriptomic and epigenomic level in our previous publication (Filarsky et al, Science 2018). The Filarsky et al. study showed that GDV1 overexpression triggers the natural (i.e. "stress-induced") AP2-G-dependent transcriptional cascade of sexual conversion and early gametocyte development and has no effect on the transcription of other genes. Using ChIP-seq experiments we showed that GDV1 binds specifically and exclusively to all HP1-enriched heterochromatic genes. However, GDV1-binding has no consequences for the bulk of heterochromatic genes. It is only the *ap2-g* locus and about five other HP1-associated gametocyte-specific genes where GDV1-binding leads to HP1 eviction and gene activation. This small number of genes is also activated during the natural process of sexual commitment and early gametocyte differentiation. Transcription of *var*, *rifin*, *stevor*, *pfmc-2tm* and all other heterochromatic genes remains unchanged. So unlike speculated by reviewer 2, GDV1 overexpression does not cause a general upregulation of all HP1/H3K9me3-marked genes. Moreover, we show that ectopic GDV1 is not expressed in NF54/iGP1 gametocytes and all subsequent stages in the mosquito vector. The same is true for NF54/iGP2 parasites, apart from the low-level expression in gametocytes that can be prevented by addition of

GlcN. Furthermore, given that the NF54/iGP1 and NF54/iGP2 gametocytes mature with normal kinetics and morphology, have unaltered sex ratios, exflagellate, are infectious to mosquitoes, produce oocysts and sporozoites that invade hepatocytes - all with similar efficiency compared to NF54 wt parasites - shows that the ectopic *gdv1* cassettes in these parasites do not affect normal parasite development throughout the life cycle. To test if transcriptional changes in some heterochromatic genes still occur in NF54/iGP gametocytes (even though there is no real evidence to suggest this may occur), we would need to perform triplicate comparative transcriptomic analyses at multiple time points across gametocytogenesis with NF54/iGP1, NF54/iGP2 and NF54 wt parasites. For the reasons outlined above, we therefore believe performing such a massive experiment is out of scope of this already very comprehensive study.

6. I don't understand why the section on 3D7 is included as an entire section of the results. Perhaps as Supplementary data, but clearly these lines are inferior to using the NF54.

>>>We moved parts of the 3D7/iGP Results section to Supplementary Note 1, and moved parts of Figure 1 to Supplementary Figure 2. The shortened Results section now serves to introduce the overall iGP concept and the sexual commitment induction protocol and SCR assay used throughout the study.

7. The authors introduce a GDV1+tag expression cassette into the *cg6* locus in both the 3D7 and NF54 parasite background lines. They explain that the chosen locus is dispensable, but what is the purpose of using this locus over the GDV1 endogenous locus? And what are the consequence of leaving a functional endogenous copy of *gdv1* in the genome? Is there any possibility that the wild-type locus is having an effect on this phenotype?

>>>Modifying the endogenous *gdv1* locus for inducible expression would be extremely challenging, if not impossible. Expression of endogenous *gdv1* is subject to a poorly understood mechanism of long non-coding antisense RNA (lnc-asRNA)-dependent control (Broadbent et al., BMC Genomics 2015; Filarsky et al., Science 2018). While an approach akin to the *ap2-g* promoter swap approach applied by the Cortes lab may have been an alternative approach in principle, it is unclear if this would overcome lnc-asRNA-dependent repression. For this reason, we decided on an inducible transgene approach and were very successful in achieving high-level gametocyte induction.

The wt *gdv1* locus is not expected to have any effect on the phenotype of the iGP lines. Endogenous GDV1 will still be expressed in the usual small subset of parasites (just like it occurs in wt parasites) and responsible for the low background SCRs observed in absence of induction of ectopic GDV1 expression. After induction of GDV1-GFP-(DD) expression, a small subset of sexually committing schizonts will express both endogenous GDV1 and ectopic GDV1-GFP-(DD) simultaneously, while most will only express ectopic GDV1-GFP-(DD).

8. From the description in the methods it seemed like the authors were culturing on 2.5 mM glucosamine long-term, how does this affect the growth of the parasites in vitro compared to a wt parasite line? It is known that certain concentrations of glucosamine are toxic in vitro and it is necessary for these effects to be quantified. In addition, the effect of addition of N-acetyl glucosamine to deplete asexual parasites during gametocyte development of *glmS*-containing parasites is unknown.

>>>Prommana and colleagues, who adapted the *glmS* riboswitch system for use in *P. falciparum*, observed no growth defect of wt parasites when using 2.5 mM GlcN (Prommana et al., PLoS One 2013). 2.5 mM GlcN also caused no growth defect on wt parasites in other studies (e.g. Aroonsri et al., Int J Parasitol 2016; Chisholm et al., PLoS One 2016), and another study even used 5 mM GlcN without observing a growth defect of wt parasites (e.g. Jankowska-Döllken et al., Sci Rep 2019). We used 2.5 mM GlcN in all experiments and observed no growth defect either.

Treatment with GlcNac is the standard approach in the field to eliminate asexual parasites from gametocyte cultures. Some laboratories use heparin treatment instead (Miao et al., Exp Parasitol 2013). To our knowledge, whether or not GlcNac (or heparin) treatment has any

effect on gametocyte development is unknown, and we are also not aware of any study investigating such possible effects. We also don't understand the reasoning why GlcNac treatment would specifically affect glmS-containing gametocytes.

9. The authors do not show that they have quantified the level of perturbation of GDV1 expression achieved with any of the modifications that were made to the locus in the NF54 lines. Evidence of induction is lacking and would be nice, although the phenotype is clear. How much more protein is in the cell? Minimally, a Western Blot that shows restoration of protein expression following Shield-1 addition or ablation of protein expression following glucosamine addition is required. In addition, due to off target cutting by Cas9, it would be beneficial to include whole genome sequencing of the modified parasite lines to verify that there are no off-target modifications. This is now standard practice.

>>>We now performed duplicate Western blots to quantify GDV1-GFP-(DD) expression levels in clones NF54/iGP1_D8 and NF54/iGP2_E9 under non-inducing and inducing conditions. As expected from the functional readouts of triplicate sexual conversion assays performed in parallel, the Western blot results demonstrate that ectopic GDV1 expression is tightly regulated and only detectable under inducing conditions (please also see our response to point 2/reviewer 1).

If off-target cutting by Cas9 indeed occurred in our lines, we would have failed to obtain viable parasites in the first place because Plasmodium parasites lack a NHEJ repair pathway to fix lethal off-target double-strand breaks in absence of homology-directed donor templates. Shinzawa et al performed WGS on CRISPR-edited *P. berghei* parasites and found no evidence for off-target effects. Similarly, we recently performed WGS of eight independent *P. falciparum* clones carrying a conditional PKA overexpression cassette in the cg6 locus (inserted using the same CRISPR/Cas9 gRNA and plasmid as used here) and found no evidence for off-target effects either (WGS deposited at the European Nucleotide Archive, accession number PRJEB40033; Hitz et al., manuscript in preparation). We hence think WGS of the iGP lines is not required and out of scope of this work.

10. In general, what happens to these parasite lines in long-term culture? From the early work of Eksi et al. it is known that the *gdv1* locus can be lost during in vitro culture. Is it possible that the incorporation of a 2nd non-selectable copy of *gdv1* (in the cg6 locus) could also be selected against in culture? Have the authors explored this idea?

>>>We have not explored this idea and cannot exclude the possibility that iGP parasites may lose the *gdv1* transgene after long term culture. However, it is standard in the gametocyte research field not to culture parasites continuously for long periods to avoid genetic alterations that lead to gametocyte-deficient phenotypes (such as mutations in/loss of the endogenous *gdv1* locus or mutations in *ap2-g*) or loss of exflagellation capacity. We never keep our parasites in continuous culture for more than 60 days and we never observed loss of the *gdv1* transgene cassette.

11. The authors use two different methods of knockdown in the NF54 background line. The first attempt using just the FKBPdd-mediated knockdown did not produce viable transfectants. So, the authors continued on to use a double knockdown method (FKBPdd+glmS) and an alternative single knockdown method (glmS). Both methods knocked-down GDV1, yet at two different levels: the protein level for FKBPdd and the RNA level for glmS. Why continue to use FKBPdd in the double knockdown system if by itself it would not produce a viable transfectant? Why not just move forward with the glmS knockdown method? Their results show that both methods could not produce more than 75% conversion, so which method would be the best option for someone wanting to use this method?

>>>Please see our response to point 1/reviewer 2.

12. The authors speculate on Page 8 that they were unable to integrate the GDV1-GFP-DD into the cg6 locus due to a deficiency in DD-dependent degradation in the NF54 strain. If this is true, can they demonstrate this with the GDV1-GFP-DD-glmS line that is successfully

integrated into the *cg6* locus? In the absence of both glucosamine and Shld-1, the authors should be able to test whether the strain 1) stabilizes GDV1 and 2) produces more gametocytes than the parental NF54 merely due to the presence of residual GDV1 that is inefficiently degraded.

>>>To address this question, we performed Western blot experiments to quantify expression of GDV1-GFP-DD in NF54/iGP1 schizonts cultured -Shld-1/+GlcN, -Shld-1/-GlcN and +Shld-1/-GlcN, and in NF54/iGP2 parasites cultured +GlcN and -GlcN. In parallel, we also quantified SCRs from the same parasite populations (new data; Figs. 2e, 2f and S5b). The SCRs under -Shield-1/-GlcN conditions were 13%, compared to 8% under fully repressive -Shield-1/+GlcN conditions (new data; Fig. 2e). These results demonstrate that the GDV1-GFP-DD fusion protein in NF54/iGP1 parasites is efficiently but not completely degraded in absence of Shield-1. 3D7/iGP parasites expressing the same GDV1-GFP-DD fusion protein showed SCRs of 3-5% under -Shield-1 conditions (Fig. 1d). This suggests that the DD-dependent degradation of GDV1-GFP-DD in NF54 parasites is indeed less efficient compared to 3D7, and this may explain why we failed to integrate the GDV1-GFP-DD version lacking the *glmS* element into the *cg6* locus in NF54 parasites. In line with this hypothesis, we observed in a recent study from our lab (Hitz et al., *Communications Biology* 2021) that while the DD-dependent depletion of the CK2 α kinase was successful in NF54 parasites, it didn't lead to the lethal phenotype that was achieved by DD-dependent degradation of the CK2 α kinase in 3D7 parasites (Tham et al., *PLoS Pathogens* 2015).

13. In this manuscript two NF54 lines are presented, both of which can transmit to mosquitoes. In my opinion, for this to truly become relevant to the malaria community at large, the authors should present a compelling reason to use one over the other, either the GDV1-GFP-DD-*glmS* or the GDV1-GFP-*glmS*.

>>>In this revised version of our manuscript, we present an in-depth characterisation of the NF4/iGP1_D8 and NF54/iGP2_E9 clones, all the way from quantifying SCRs to the production of hepatocyte infection (new data; Figs. 2e-g, 3a-c, S5b, S5c, S6a-d, S7a-c). Based on these results, we consider both lines as highly valuable tools for future research. We state this explicitly in the Discussion section, Fig. 6 and also provide a new Table S2 summarising their specific properties and making recommendations for their potential use in the different fields of basic and applied transmission stage research.

14. Although the authors state in the discussion that they do not know why they fail to get 100% conversion, this should be explored further. Again, is this perhaps due to the presence of the endogenous *Gdv1* protein?

>>>In our opinion, expression of the endogenous GDV1 protein cannot explain the failure to obtain 100% sexual conversion. One possibility is that the *gdv1* lnc-asRNA from the endogenous locus may interfere with expression of the GDV1-GFP-(DD) transgene in some parasites. Another possibility is that the threshold level of GDV1-GFP-(DD) expression required to achieve sexual conversion (Filarsky et al., *Science* 2018) may not be reached in all parasites in the population due to natural fluctuations in protein expression between individual cells. Solving the underlying reason(s) experimentally, however, would require a plethora of additional cell lines and experiments that would go far beyond the scope of this work. We would also like to mention that none of the previously published experimental cell lines for gametocyte induction achieved SCRs of 100% (e.g. Brancucci et al., *Cell Host Microbe* 2014; Filarsky et al., *Science* 2018; Llorca-Batlle et al., 2020), and again the underlying reasons remain unknown. We therefore removed this sentence from the Discussion.

Figure Comments:

15. Figure 1b is not necessary as a main panel and can be moved to the Supplementary data.

>We disagree with the reviewer on this point. Figure 1b introduces the culture protocol used to induce sexual commitment and gametocytogenesis via induction of ectopic GDV1 expression, as well as the time point for anti-Pfs16 IFAs to quantify SCRs. These protocols

have been used throughout the study and we believe it is important to be visualised right at the beginning of the Results section.

16. Figure 2: The description of the conversion rate assay is difficult to interpret. It is not clear how these conversion rates are quantified? Are these rates a function of gametocytemia/ rings or a percentage of the total culture on day 2?

>>>The SCR assay is illustrated in Fig. 1b and explained in the corresponding legend. We now also modified the Methods section accordingly to better explain how SCRs were calculated: "To quantify SCRs, parasites were methanol-fixed at 36-44 hpi in generation 2 (day 2 of gametocytogenesis) and α -Pfs16 IFAs combined with DAPI staining were performed. SCRs were determined as the proportion of early stage I gametocytes (DAPI-positive/Pfs16-positive) among all infected RBCs (DAPI-positive)."

17. Pfs16 is also expressed in asexual stages and the time of sampling was too early in development for the line to be enriched for gametocytes. From the IFA it would seem that the authors are counting schizonts that are expressing Pfs16 as gametocytes, however, there are no data showing all Pfs16-expressing asexual parasites will develop into gametocytes, which would result in an overestimation of gametocytemia.

>>>We respectfully disagree with the reviewer also on this point. The time point of parasite sampling for anti-Pfs16 IFAs was not too early. Pfs16 is a highly gametocyte-specific protein expressed in early stage I gametocytes as early as 20 hours post-invasion (Bruce et al., Mol Biochem Parasitol 1994; Bancells et al., Nature Microbiology 2018). While pfs16 mRNA, or pfs16 promoter-driven GFP expression, has been detected at low levels in asexual parasites (Eksi et al., Mol Biochem Parasitol 2008), the native Pfs16 protein is only expressed in gametocytes and not in asexual parasites (Lanfrancotti et al., Exp Parasitol 2007; Bruce et al., Mol Biochem Parasitol 1994; Bancells et al., Nature Microbiology 2018).

We performed the anti-Pfs16 IFAs 36-44 hours post-invasion in the generation directly following the cycle of induction of sexual commitment through GDV1-GFP-(DD) expression. We specifically chose this time point to obtain the most accurate readout for SCRs we can think of. At this stage (i) the asexual progeny is still in generation 2 and has not proliferated further; (ii) the sexual progeny are early stage I gametocytes and express Pfs16; and (iii) the asexual parasites are Pfs16-negative and are generation 2 schizonts containing multiple nuclei (in contrast to stage I gametocytes that carry a single nucleus). Together, these parameters allow for the unambiguous identification of early sexual stage parasites in the direct progeny of GDV1-induced generation 1 schizonts, and hence for an accurate quantification of SCRs: i.e. the proportion of Pfs16-positive/DAPI-positive cells (stage I gametocytes) among all DAPI-positive cells (stage I gametocytes PLUS asexual schizonts). The representative IFA images shown in Fig. 1c illustrate this impressively by showing that all Pfs16-positive parasites carry a single nucleus, whereas the Pfs16-negative cell is identified as a schizont based on the presence of multiple nuclei. Please also note that this assay is standard in the field and not only used in our lab.

Furthermore, comparing the SCRs determined by Pfs16 IFAs on day 2 of gametocytogenesis (Fig. 2b) are very comparable to the proportion of stage III gametocytaemia determined on day 6 compared to the total asexual/sexual ring stage parasitaemia determined on day one (Fig. 2d), showing that Pfs16-positive cells identified by IFA on day 2 of gametocytogenesis develop further into more mature stages.

18. The experiment would be much easier to interpret with a WT NF54 control to relate back to. Without knowing the basal commitment rate it's difficult to relate the scope of improvement in the transgenic line.

>>>We performed this control experiment, revealing a basal commitment rate of 2% for our NF54 wt control strain (new data; Fig. 2e).

19. Why are the conversion rates in the clonal lines lower? Were the experiments in the clonal line replicated? According to the figure legend, I think not, but if so, please show the data. If not, please provide additional replicates. Repetition would show whether it is a

statistical oddity or a real biological effect? There are often big variations in commitment between inductions, it is not convincing enough to have a single experiment as a defining result for the paper. Since the non-clonal lines contain parasites with the episomal plasmids/concatamerized plasmids, is the hypothesis that these parasites result in better conversion? It is odd that a mixed culture would produce a better result reproducibly?

>>>We thank reviewer 2 for this comment. The induction experiments with the clonal lines were initially performed only once on a single day as we thought it would be sufficient to demonstrate that the induction system is still functional in three independent clones each. We now selected one clonal line each and performed three additional induction experiments each (new data; Fig. 2e). NF54/iGP_D8 shows a mean SCR of 63%, NF54/iGP_E9 shows a mean SCR of 73%. Hence, the NF54/iGP2_E9 clone has similar SCRs as the mother line. The NF54/iGP1_D8 clone has indeed a slightly lower SCRs compared to the mother line. We cannot explain why but believe this may be due to clonal variation in commitment capacity, which is frequently observed also in wt parasites. However, we still consider SCRs of 63% remarkable. Regarding concatamerised plasmids, we would like to note that they were detected only in the NF54/iGP1 mother line (Fig. S3).

20. Does the variability in conversion between strains in Figures 2b and 2d arise from differences in expression of Gdv1 from the endogenous *gdv1* locus? The authors should test this.

>>>We believe there may be a misunderstanding here. There is no marked variability in SCRs between the strains shown in Figs. 2b and 2d (now Fig. 2b). We would nevertheless like to comment on the reviewer's hypothesis: testing if differences in SCRs are due to differences in endogenous GDV1 expression is very difficult due to the co-expression of the regulatory *Inc-asRNA* (Filarsky et al., Science 2018) and the lack of antibodies against endogenous GDV1.

21. Why is the mother line tested in Figure 2b and 2d? What is the relevance? And in the figure legend the term "progeny of the NF54/iGP1 mother line" should either be "progeny clones" or just "clones".

>>>The relevance of testing the mother lines is to show that the systems work in the first place. Furthermore, some researchers prefer to use uncloned lines, some prefer using cloned lines and sometimes clones lose certain phenotypes by chance, so we think it is important to show these data. In the figure legend, progeny refers to the offspring produced upon reinvasion so we think it is fine to say progeny. In the revised version of the manuscript we put clear emphasis on the clonal lines, but we still also performed SMFAs with the mother lines and think it is valuable to present these data as well.

22. In Figure 3, it appears that the DD-containing parasite is less effective at producing oocysts at all feeding days. Does this imply perhaps that there may be a secondary role for GDV1 that is not able to function in the absence of Shld-1? While the strain lacking the DD (only regulated by the *glmS*) continues to express the protein, allowing for proper function in many?, some?, all? cells.

>>>To date, it is unknown if GDV1 has any function in any stages other than sexually committed trophozoites/schizonts. If GDV1 has any role in oocysts, this role would be carried out by endogenous GDV1 that would be expressed equally in NF54/iGP1, NF54/iGP2 and NF54 wt oocysts. We did not detect ectopic GDV1 expression by live cell fluorescence microscopy and IFA in oocysts in both iGP lines. It is therefore highly unlikely that the ectopic GDV1 expression cassettes have any effect on oocyst intensities.

The DD-containing NF54/iGP1 mother line and clone D8 produced similar numbers of oocysts like NF54 wt parasites, fewer than the NF54/iGP2_E9 clone and more than the NF54/iGP2 mother line (new data; Figs. 3 and S7). Results from SMFAs are highly variable and the differences observed are well within the normal biological variation seen in all mosquito feeding experiments.

23. Also Figure 3: It would be vastly preferred if the authors had performed their own NF54 mosquito infection control with the strain used to transfect the plasmids into if possible.
>>>We now performed these control experiments with NF54 wt gametocytes (new data; Fig. 3a and 3b).

24. For Figure 3 were the same number of cells used? The methods say approximately 3%, but were they set up to match as closely as possible?
>>>With the new SMFA experiments conducted for this revision, the gametocytaemia was in the range of 3-10% for the 21 different stage V cultures that were fed to mosquitoes (indicated in the Methods section). It is standard practice to not adjust gametocytaemias prior to performing the SMFAs because there is no evidence that gametocytaemia has a significant effect on mosquito infection rates in these experiments.

25. Figure 4a can be reduced in size. Perhaps the microscopy can be side-by-side rather than stacked on top for asexual and gametocyte stages?
>>>We prefer to stay with the current display of these IFA results as we consider it important to show single channel images (including DIC to appreciate size and position of parasites inside the RBC) as well as merged images and zoom-in views to appreciate the localisation of NPCs in relation to bulk chromatin.

26. Figure 4&5: Tagging a nucleoporin to characterize the nuclear boundaries of the gametocyte is a nice addition as proof-of principle for making additional modifications in this line. However, without doing the same in a control strain, that does not modify a gene involved in transcriptional regulation within gametocytes, it is difficult to conclude from these data that the nuclear structure of the gametocyte is not modified by the overexpression of GDV1.
>>>We can't understand this argument as there is no evidence to suggest that the ectopic GDV1 cassette would modify the structure of parasite nuclei. Full overexpression of GDV1-GFP doesn't affect nuclear structure in asexual parasites, Full overexpression of GDV1-GFP has only a tiny effect on gene expression by activating ap2-g and about five other early gametocyte genes (Filarsky et al., 2018). GDV1 is not known to regulate transcription in gametocytes. Ectopic GDV1-GFP is hardly expressed in NF54/iGP2 gametocytes and undetectable in NF54/iGP1 gametocytes (Fig. S5c). Both NF54/iGP2 and NF54/iGP1 gametocytes are healthy and viable and infectious to mosquitoes (Fig. 3). Furthermore, TEM studies of gametocytes already demonstrated irregular nuclear shapes in wt gametocytes (e.g. Sinden et al., Proc. R. Soc. Lond B Biol. Sci 1978; Sinden, Parasitology 1982). We therefore believe that tagging NUP313 in NF54 wt parasites and repeating microscopy is unnecessary and out of scope of this work.

27. Figure 6 is a nice overview, but there is nothing in this work demonstrating that oocysts can go on to produce viable sporozoites that can infect hepatocytes. If such data exist, please add them as they would strengthen the manuscript. Otherwise, consider shortening the figure at the oocyst stage.

>>>As outlined above, we now performed these experiments and show that the NF54/iGP1 and NF54/iGP2 parasites produce viable sporozoites that can infect hepatocytes (new data; Figs. 3b and 3c). We therefore kept the original Fig 6.

Minor Comments:

28. Page 6: Change "...non-proliferative cells and thus rapidly overgrown..." to "...non-proliferative cells and thus are rapidly overgrown..."

>>>We changed this sentence accordingly.

29. Page 8: Change "...the sexual ring stage progeny was exposed to 50 nM ... to eliminate asexual parasites and was thereafter..." to "...the sexual ring stage progeny were exposed to

50 nM ... to eliminate asexual parasites and were thereafter..."

>>>We changed this sentence accordingly.

30. Page 11: should be nuclear lamina, currently "lamin".

>>>In this sentence "lamin" refers to the protein lamin, not the lamina structure.

31. Page 11: Reference formatted incorrectly (Field and Rout, 2019.

>>>We corrected this error.

32. Page 13: Discussion: "Fourth, gametocyte synchronicity and yield are highly reproducible between experiments; starting with approximately 0.75×10^7 iGP ring stage parasites in the commitment cycle (5% hematocrit, 1.5% parasitemia) routinely delivers up to 2×10^8 stage V gametocytes per 10 ml of culture volume in a single induction experiment..." The data that led up to this observation would be much better served with quantification in the manuscript than merely described in the discussion.

>>>This was indeed quantified in the Results section (initial Fig. 2f, now Fig. 2d). The calculation in the Discussion section/Fig. 6 simply puts the SCRs and gametocytemias obtained from typical induction experiments starting with a defined culture volume at defined parasitaemia and haematocrit into absolute numbers.

Reviewer #3 (Remarks to the Author):

The development of gametocytes during the blood stage of the malaria parasite's life cycle is crucial for malaria transmission. It has been difficult to study the process of gametocytogenesis because not all strains produce gametocytes and the conditions to trigger gametocytogenesis are not understood. However, significant progress has been made in our understanding of the genetic regulation of gametocytogenesis in malaria parasites in recent years. The authors have previously made observations that the nuclear protein, gametocyte development 1 (GDV1), is essential for gametocytogenesis in *P. falciparum*. In this manuscript, the authors have used this observation to develop a method to develop a transgenic *P. falciparum* line in which expression of GDV can be switched on in blood stage cultures to trigger the production of gametocytes. This approach provides a reliable and efficient production of synchronized *P. falciparum* gametocytes providing a useful tool to produce gametocytes and study the process of gametocytogenesis. A similar approach in which PfAP2-G was conditionally expressed in a transgenic *P. falciparum* line to produce gametocytes has been previously described (Llora-Battle et al., *Sc Adv* 2020). However, male gametocytes did not exflagellate and therefore mosquito transmission was not achieved. Here, although the authors do not specifically analyse formation of male and female gametocytes and the competence of latter to exflagellate, they do demonstrate the ability of induced gametocytes to infect mosquitoes and form oocysts.

>>>We thank reviewer 3 for her/his appreciation of our work.

The authors should address the following comments:

1. The authors do not actually demonstrate changes in levels of GDV1 in presence/absence of Shield 1 and GlcN as appropriate in the different transgenic lines. The authors should perform Western blot analysis to demonstrate changes in GDV1 levels following addition/removal of Shield 1 and GlcN.

>>>We now performed these Western blot experiments and show that GDV1-GFP(-DD) expression is only detectable under culture conditions that induce ectopic GDV1 expression (new data; Fig. 2f and S5b) (please also see our responses to point 2/reviewer 1 and points 2, 9, 12/reviewer 2).

2. Is there any difference in male/female gametocyte ratios following induction of gametocytogenesis by expression of GDV1? Is the efficiency of exflagellation of male

gametocytes following induction of gametocytes by expression of GDV1 in transgenic line similar to that in wt strain? Similarity in such phenotypes will confirm that this model does in fact reflect the normal process of gametocytogenesis in wt parasites.

>>>We performed IFAs using antibodies against the female-specific protein Pfg377 to quantify sex ratios in NF54/iGP2 gametocytes (new data; Figs. 2g and S6a) and NF54/iGP1 gametocytes (new data; Fig. S6b). The results show that sex ratios do not differ between gametocyte populations obtained via ectopic GDV1 expression or stress-induced sexual conversion.

We also quantified exflagellation and observed no difference in exflagellation efficiency between NF54/iGP1, NF54/iGP2 and NF54 wt gametocytes (new data; Fig. S6c).

3. The authors further genetically manipulate the transgenic gametocyte producing lines to tag a nucleoporin (PfNUC313) with a 'scarlet' fluorescent protein to follow the distribution of nuclear pores during the process of gametocytogenesis in gametocytes at different stages. While they are successful at doing so, these data are purely descriptive. It is not clear what question or hypothesis these experiments were trying to address.

>>>As explained in the manuscript, we wanted to demonstrate that NF54/iGP parasites can easily be genetically modified a second time because the capacity to produce large numbers of genetically engineered gametocytes offers unprecedented possibilities for basic and applied research. To demonstrate this capacity we could of course have targeted many other genes. However, our lab has a strong interest in studying nuclear biology of malaria parasites, and with this new iGP tool in hand we can finally begin widening this research to gametocytes. As a first step, we decided to tag PfNUP313 as a marker for the nuclear envelope to revisit intriguing observation made over four decades ago by TEM studies, suggesting profound changes in nuclear shape and size in gametocytes. Furthermore, we were interested in learning whether the number and distribution of nuclear pores in gametocytes change as dramatically as observed in asexual parasites. Our results successfully addressed these questions and, while being purely descriptive at this stage, delivered convincing results and a strong incentive to study the role of nuclear architecture and genome organisation in gametocyte biology. We now tried to explain better why we decided to visualise nuclear pores in gametocytes

Reviewer comments, second round –

Reviewer #2 (Remarks to the Author):

I commend the authors for the comprehensive revisions and additional data, in particular the exciting new liver stage data.

I have two remaining questions:

First, do the authors believe that the DD-FKBP tag adding any value to the GDV1 knockdown?

Second, while whole genome sequencing may be beyond the scope of the requirements for publication, it would seem to me that if, as the authors propose, these lines will become standards in the field for large-scale gametocyte experimentation, it would be advantageous to ensure that they are indeed free of other significant mutations.

Reviewer #3 (Remarks to the Author):

The authors have addressed my comments adequately.

Author's response letter (NCOMMS-20-39775A)

Reviewer #2 (Remarks to the Author):

I commend the authors for the comprehensive revisions and additional data, in particular the exciting new liver stage data.

>>>We thank reviewer 2.

I have two remaining questions:

First, do the authors believe that the DD-FKBP tag adding any value to the GDV1 knockdown?

>>>According to our data, the DD tag doesn't improve the knockdown efficiency beyond what is already achieved by the glmS riboswitch element. However, the DD tag secures efficient knockdown of GDV1 expression under conditions where glucosamine should be avoided or can't be used, so the NF54/iGP1 lines/clones are still useful.

Second, while whole genome sequencing may be beyond the scope of the requirements for publication, it would seem to me that if, as the authors propose, these lines will become standards in the field for large-scale gametocyte experimentation, it would be advantageous to ensure that they are indeed free of other significant mutations.

>>>We agree with reviewer 2 on this point and may still perform WGS of the NF54/iGP clones in the future.

Reviewer #3 (Remarks to the Author):

The authors have addressed my comments adequately.

>>>We thank reviewer 3.